# Coupling between motor cortex and striatum increases during sleep over long-term skill learning

**Stefan M Lemke[1,2,3,4], Dhakshin S Ramanathan[5], David Darevksy[2,3], Daniel Egert[3], Joshua D Berke[3,6], Karunesh Ganguly[2,3,6]***

[1]Neuroscience Graduate Program, University of California, San Francisco, San Francisco, United States; [2]Neurology Service, San Francisco Veterans Affairs Medical Center, San Francisco, United States; [3]Department of Neurology, University of California, San Francisco, San Francisco, United States; [4]Istituto Italiano di Tecnologia, Rovereto, Italy; [5]Department of Psychiatry, University of California, San Diego, San Diego, United States; [6]Weill Institute for Neurosciences and Kavli Institute for Fundamental Neuroscience, University of California, San Francisco, San Francisco, United States

**Abstract** The strength of cortical connectivity to the striatum influences the balance between behavioral variability and stability. Learning to consistently produce a skilled action requires plasticity in corticostriatal connectivity associated with repeated training of the action. However, it remains unknown whether such corticostriatal plasticity occurs during training itself or 'offline' during time away from training, such as sleep. Here, we monitor the corticostriatal network throughout long-term skill learning in rats and find that non-rapid-eye-movement (NREM) sleep is a relevant period for corticostriatal plasticity. We first show that the offline activation of striatal NMDA receptors is required for skill learning. We then show that corticostriatal functional connectivity increases offline, coupled to emerging consistent skilled movements, and coupled cross-area neural dynamics. We then identify NREM sleep spindles as uniquely poised to mediate corticostriatal plasticity, through interactions with slow oscillations. Our results provide evidence that sleep shapes cross-area coupling required for skill learning.

*For correspondence:
karunesh.ganguly@ucsf.edu

**Competing interests:** The authors declare that no competing interests exist.

## Introduction

Cortical and basal ganglia circuits regulate behavioral variability, as evidenced in habit development (*Gremel et al., 2016*; *O'Hare et al., 2016*; *Rueda-Orozco and Robbe, 2015*; *Malvaez and Wassum, 2018*; *Lipton et al., 2019*; *Yin and Knowlton, 2006*), skill learning (*Santos et al., 2015*; *Kupferschmidt et al., 2017*; *Koralek et al., 2012*; *Yin et al., 2009*), as well as the pathophysiology of neuropsychiatric disorders such as obsessive-compulsive disorder and autism spectrum disorder (*Vicente et al., 2020*; *Shepherd, 2013*). In the case of skill learning, the ability to consistently produce a skilled action is accompanied by emerging coordinated neural activity across the motor cortex and striatum during action execution (*Santos et al., 2015*; *Lemke et al., 2019*; *Koralek et al., 2013*). Skill learning has also been associated with striatal NMDA receptor activation (*Santos et al., 2015*; *Koralek et al., 2012*; *Jin and Costa, 2010*; *Dang et al., 2006*), suggesting that the activity-dependent potentiation of cortical inputs to the striatum may be required (*Calabresi et al., 1992*; *Charpier and Deniau, 1997*). However, little is known about the specific activity patterns that may drive corticostriatal plasticity, or when they occur, during skill learning.

One intriguing possibility is that neural activity patterns during 'offline' periods, or time away from training such as sleep, play a central role in driving corticostriatal plasticity during skill learning.

This possibility is motivated by evidence that 'reactivations' of training-related neural activity patterns during sleep promote motor skill learning (*Yang et al., 2014*; *Ramanathan et al., 2015*; *Gulati et al., 2014*; *Kim et al., 2019*). Moreover, it has been proposed that both cortical and subcortical brain areas are engaged during the sleep-dependent consolidation of motor skills (*Vahdat et al., 2017*; *Boutin et al., 2018*; *Doyon et al., 2018*; *Doyon and Benali, 2005*). However, the specific activity patterns that may impact cross-area connectivity during sleep, or how such sleep-dependent plasticity may impact network activity during subsequent awake behavior, remains unknown.

Here, we monitor the corticostriatal network throughout reach-to-grasp skill learning in rats. We establish that the coupling between motor cortex and striatum increases offline during skill learning, and identify sleep spindles during non-rapid-eye-movement sleep (NREM) as uniquely poised to mediate such plasticity, through interactions with slow oscillations (SOs). We first show that blocking striatal NMDA receptor activation during offline periods following training disrupts the emergence of a consistent skilled action. We then show that corticostriatal functional connectivity increases offline, rather than during training itself, and that such offline plasticity tracks increased movement consistency and emerging coupled cross-area neural dynamics during action execution. We then demonstrate that sleep spindles in NREM uniquely facilitate corticostriatal network transmission and link the modulation of M1 and DLS neurons during sleep spindles following training to the preservation of corticostriatal functional connectivity. Finally, we provide evidence that the temporal proximity between sleep spindles and SOs influences the impact that sleep spindles have on the corticostriatal network. These results provide evidence that NREM rhythms play a role in strengthening cross-area connectivity in the corticostriatal network during skill learning.

## Results

We implanted six adult rats with either microwire electrode arrays (*n*=5) or custom-built high-density silicon probes (*Egert et al., 2020*) (*n*=1) in the primary motor cortex (M1) and the dorsolateral striatum (DLS), which receives the majority of M1 projections to the striatum (*Figure 1a*; *Figure 1—figure supplement 1*; *Aoki et al., 2019*). Neural activity in both regions was simultaneously monitored as rats underwent long-term reach-to-grasp skill training (range: 5–14 days). On each day, rats were placed in a custom-built behavioral box (*Wong et al., 2015*) and neural activity was recorded during a 2–3 hr pre-training period (consisting of both sleep and wake periods), a 100–150 trial training period, and a second 2–3 hr post-training period (*Figure 1b*; pre-training period length: 154.1±6.1 min, post-training period length: 159.4±5.4 min, mean± SEM, *n*=56 days). Rats learned a reach-to-grasp task which involved reaching through a small window in the behavioral box to grasp and retrieve a food pellet. During pre- and post-training periods, behavioral states—wake and sleep (NREM and REM) —were classified using standard methods based on cortical local field potential (LFP) power and movement measured from video or electromyography (EMG) activity (*Watson et al., 2016*).

### Blocking offline striatal NMDA receptor activation disrupts skill learning

With repeated days of training on the reach-to-grasp task, animals developed both a consistent spatial reaching trajectory and temporal velocity profile (*Figure 1c*). To measure learning, we quantified day-to-day changes in the velocity profile, as this captured the combination of individual movements (i.e., reach toward pellet, time spent interacting with pellet, and retraction with pellet) into a consistent skilled action and was less constrained by the task than the spatial reaching trajectory. Comparing the trial-averaged velocity profile on each day of training to the trial-averaged velocity profile on the last day of training, which served as a learned 'template,' revealed that a consistent day-to-day velocity profile emerged within the first 8 days of training (*Figure 1d*; *Figure 1—figure supplement 2*). Single-trial peak reaching velocity also generally increased across training days, correlated to the consistency of the velocity profile (*Figure 1—figure supplement 3*; *r*=0.55, *P*=$5\times10^{-5}$, Pearson's *r*), consistent with previous work suggesting that movement speed is a relevant aspect of skill learning (*Lemke et al., 2019*; *Hikosaka et al., 2013*).

We next tested whether disrupting offline striatal activity and plasticity impacted learning. We trained a new cohort of animals (*n*=six rats) for 10 days, infusing either 1 µl of NMDA receptor

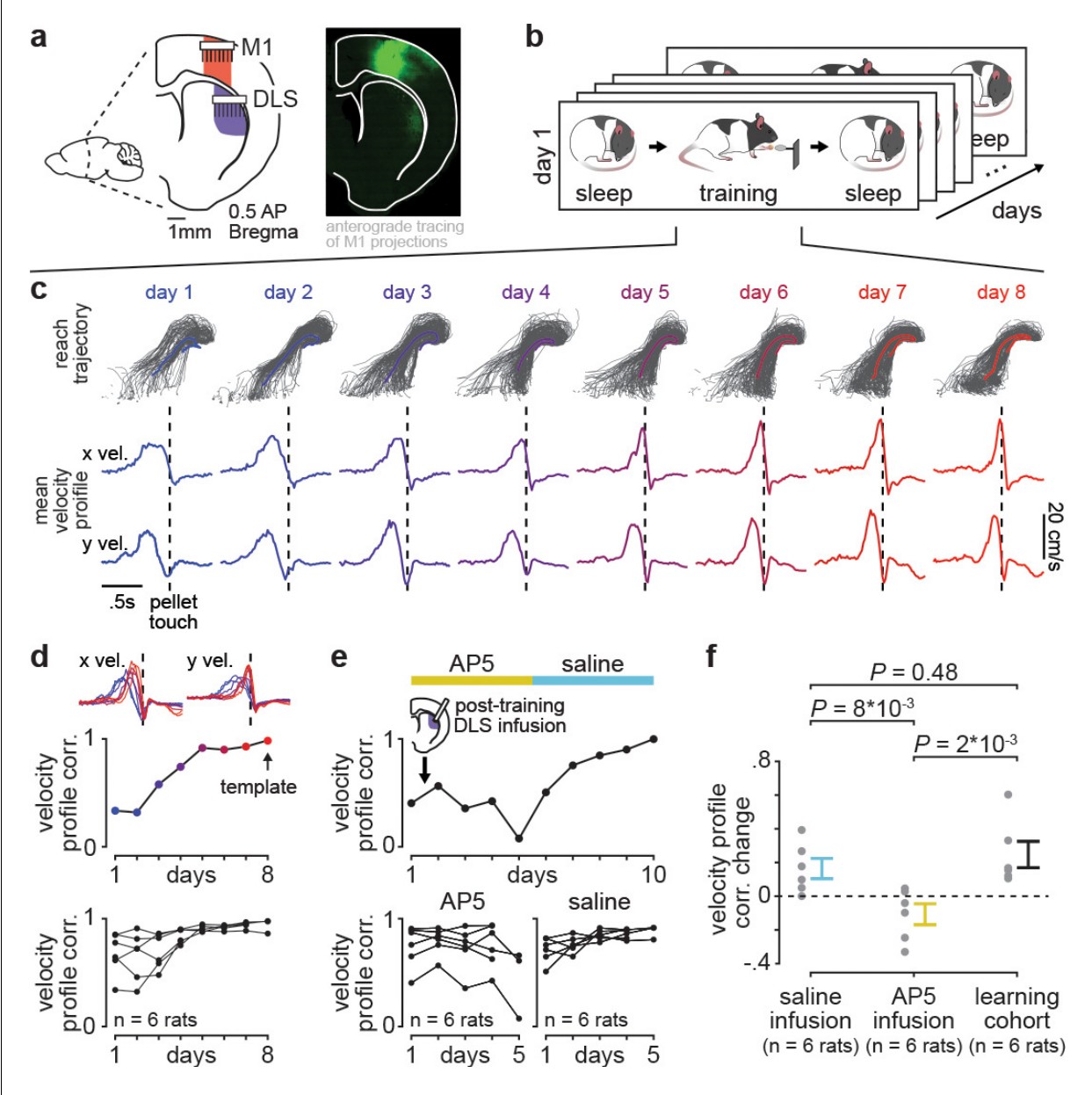

**Figure 1.** Blocking offline striatal NMDA receptor activation disrupts skill learning. (a) Schematic of recording locations in primary motor cortex (M1) and dorsolateral striatum (DLS) and anterograde tracing of M1 projections showing direct input to the DLS. (b) Schematic of each day's recording periods throughout long-term training. (c) Spatial reaching trajectories (individual trials in gray overlaid with mean trajectory in color) and mean reaching velocity profiles on each day of training in example animal. (d) Correlation between each day's mean velocity profile and the final day's mean velocity profile (average of x and y dimensions) for each day of training in example animal (top) and across animals (bottom; individual animals as black lines; last day of training, which served as template, is excluded in each animal). (e) Correlation between each day's mean velocity profile and the final day's mean velocity profile for example animal with post-training DLS infusions of AP5 or saline (top) and across animals (bottom; individual animals as black lines; last day of training, which served as template, is excluded in each animal). (f) Comparison of total change in velocity profile correlation across days with either post-training saline infusions, post-training AP5 infusions, or no infusions as in learning cohort animals (individual animals as gray dots and mean± SEM across animals in color; last day of training, which served as template, is excluded from calculations).

The online version of this article includes the following figure supplement(s) for figure 1:

**Figure supplement 1.** Electrophysiology recordings from M1 and DLS.

**Figure supplement 2.** Learning curves in individual animals.

**Figure supplement 3.** Reaching velocity across learning in individual animals.

**Figure supplement 4.** Velocity profile correlation across days in individual animals receiving post-training DLS infusions of AP5 or saline.

**Figure supplement 5.** Day-to-day changes in single-trial peak reaching velocity with post-training DLS infusions of AP5 or saline.

antagonist AP5 (5 µg/µl) or saline into DLS immediately after training on each day (*Figure 1e*). This revealed that offline striatal NMDA activation was important for skill learning, as day-to-day changes in velocity profile correlation were significantly decreased with AP5 infusions, compared to saline infusions or changes observed in the learning cohort (*Figure 1f*; *Figure 1—figure supplement 4*; *n*=six rats with AP5 infusions, −0.11±0.06 total correlation value change, mean± SEM, *n*=six rats with saline infusions, 0.16±0.06 total correlation value change, *n*=six rats in learning cohort, 0.25±0.08 total correlation value change; AP5 infusions vs. saline infusions: $P=8\times10^{-3}$, Wilcoxon rank-sum test, AP5 infusions vs. learning cohort: $P=0.48$, Wilcoxon rank-sum test, saline infusions vs. learning cohort: $P=2\times10^{-3}$, Wilcoxon rank-sum test; for all animals, the last day of training which served as template was excluded from total correlation value change calculation). Day-to-day changes in single-trial peak reaching velocity for animals receiving either AP5 or saline infusions followed a similar trend (*Figure 1—figure supplement 5*).

## Corticostriatal functional connectivity increases offline during skill learning

Given the importance of offline striatal NMDA receptor activation for skill learning, we next examined whether changes in corticostriatal functional connectivity occurred during training itself or offline, between daily training sessions. To measure functional connectivity we calculated 4–8 Hz LFP coherence across each M1 and DLS electrode during each pre- and post-training period throughout learning, as LFP signals in the theta frequency band have been previously shown to reflect corticostriatal spiking activity (*Lemke et al., 2019*; *Koralek et al., 2013*; *Thorn and Graybiel, 2014*). We calculated coherence specifically during NREM to establish a consistent measure of functional connectivity across days (*Figure 2a*). While LFP signals are generally more stable across days than single-unit spiking activity (*Flint et al., 2016*), there are significant challenges in interpreting LFP signals from non-laminar structures such as the striatum (*Tanaka and Nakamura, 2019*; *Buzsáki et al., 2012*), including the influence of non-local signals volume conducted from cortex (*Lalla et al., 2017*). To address these issues, we first locally referenced signals, in M1 and DLS separately, to decrease common noise and minimize volume conduction within each region (*Lemke et al., 2019*). This resulted in a phase difference between M1 and DLS 4–8 Hz LFP signals during NREM that was inconsistent with volume conduction (*Figure 2—figure supplement 1*). We next confirmed that 4–8 Hz LFP coherence between M1 and DLS electrodes was correlated to a separate measure of functional connectivity, calculated independently of DLS LFP: the phase locking of DLS units to M1 LFP. We calculated the entrainment of DLS units to 4–8 Hz M1 LFP signals for each M1 electrode (*Figure 2b*) and compared it to the mean 4–8 Hz LFP coherence between that M1 electrode and all simultaneously recorded DLS LFP signals (*Figure 2c and d*). We found that M1 electrodes with high LFP coherence with DLS electrodes also entrained DLS units to a greater degree than M1 electrodes with low LFP coherence with DLS electrodes (*Figure 2e and f*). Finally, we sought to test the relevance of specifically 4–8 Hz LFP coherence, versus other frequency bands. We found a significant relationship between the emergence of a consistent skilled action and mean LFP coherence measured during the pre-training period for frequencies between ~5 and 11 Hz (*Figure 2—figure supplement 2*), indicating that offline LFP coherence in the theta frequency range uniquely reflects network changes relevant to learning.

Having established LFP coherence as a measure of corticostriatal functional connectivity, we next examined whether coherence increased during training periods (online) or offline, between training periods (*Figure 3a*). Across animals, 35% of M1 and DLS electrode pairs increased in 4–8 Hz LFP coherence from the first to last day of training (36% did not change, 29% decreased; increase or decrease defined as a change in coherence of at least, .025 from first to last day of training). Across the population of electrode pairs with learning-related increases, 4–8 Hz LFP coherence increased predominantly offline, that is, between each day's post-training period and the next day's pre-training period, rather than online during training, that is, between pre- and post-training periods on the same day (*Figure 3a–c*; *Figure 3—figure supplement 1*). Across animals, mean changes in 4–8 Hz LFP coherence occurring online were not significantly different than zero, while changes occurring offline were skewed positive (*Figure 3d*; online LFP coherence changes: $t(1413)=-0.34$, $P=0.73$, offline LFP coherence changes: $t(1413)=23.4$, $P=7\times10^{-103}$ one-sample *t*-test). Notably, there was a close relationship between 4–8 Hz LFP coherence measured during the pre-training period on each day and consistency in skilled action execution during the subsequent training period (*Figure 3e*;

$r$=0.73, $P$=5×10⁻¹⁰, Pearson's $r$), indicating that offline increases in corticostriatal functional connectivity closely tracked skill learning. This relationship remained significant when taking into account single-trial peak reaching velocity ($r$=0.62; $P$=1×10⁻⁴, Pearson partial correlation coefficient).

## Offline increases in corticostriatal functional connectivity predict emergence of cross-area neural dynamics during subsequent skill execution

We next examined whether offline increases in corticostriatal functional connectivity impacted corticostriatal network activity during subsequent training periods. To measure cross-area neural dynamics, we extracted low-dimensional neural trajectory representations of DLS spiking activity during the reaching action using principal component analysis (PCA) and examined the evolution of how well M1 spiking activity could predict DLS neural trajectories over the course of learning (*Figure 4a*). To determine this predictive ability, on each day of training we fit a linear regression model to predict DLS neural trajectories (top three PCs, with a separate model fit for each component) from spiking activity in M1 and then measured the correlation between the predicted and real DLS neural trajectories. We found that the ability to predict DLS neural trajectories during execution of the reaching action increased with training, while the ability to predict the trajectory representations of DLS activity during a baseline, non-reaching, period did not significantly change (*Figure 4b*; action execution: 0.18±0.05 Pearson's $r$ on first 3 training days (average of three correlation values corresponding to

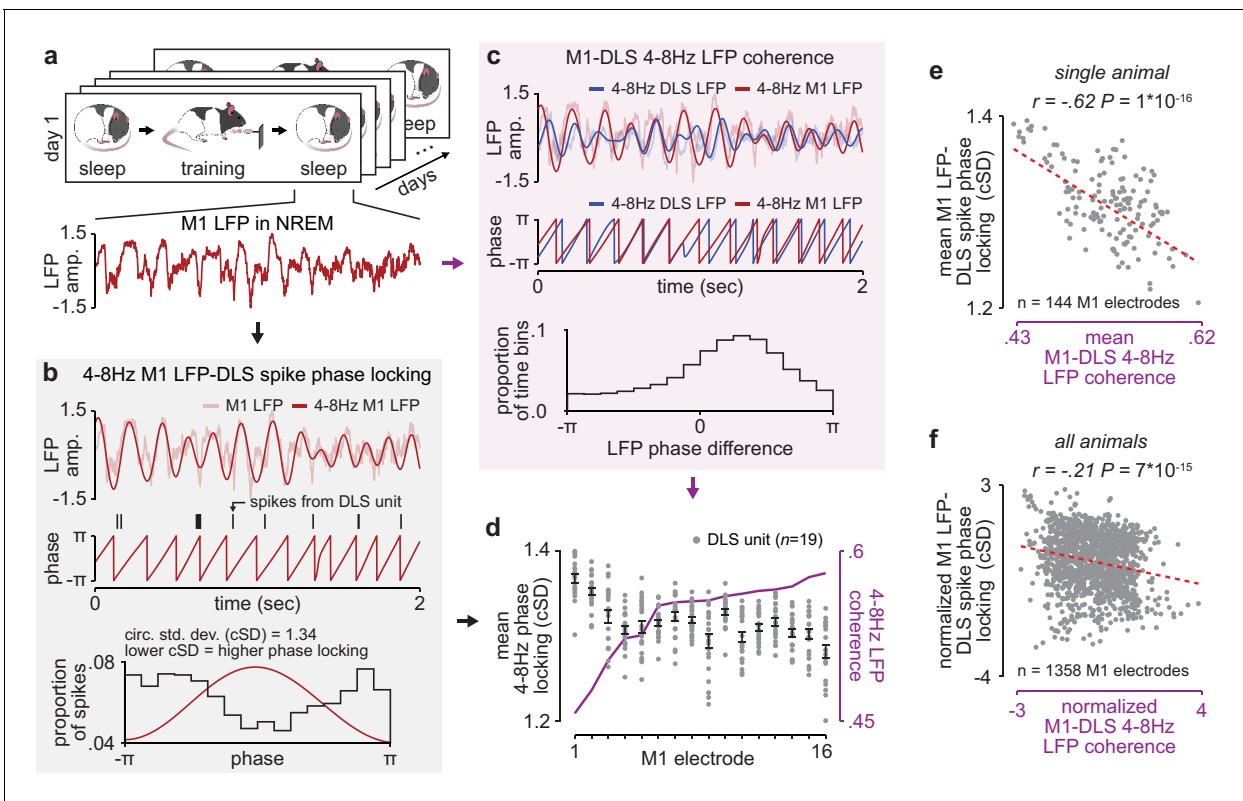

**Figure 2.** NREM M1-DLS 4–8 Hz LFP coherence reflects M1 LFP-DLS spike phase locking. (a) Example snippet of M1 LFP during NREM. (b) Example computation of M1 LFP-DLS spike phase locking. Lower circular standard deviation (cSD) is equivalent to greater phase locking. (c) Phase difference between M1 and DLS 4–8 Hz LFP signals for example electrode pair with high coherence. (d) Relationship between mean M1 LFP-DLS spike phase locking and 4–8 Hz M1-DLS LFP coherence for all M1 electrodes in example animal on example day. (e) Scatterplot between mean M1 LFP-DLS spike phase locking and 4–8 Hz M1-DLS LFP coherence for M1 electrodes across all days in example animal. (f) Same as (e) for M1 electrodes across all animals. M1, primary motor cortex, DLS, dorsolateral striatum; LFP, local field potential; NREM, non-rapid-eye-movement sleep.

The online version of this article includes the following figure supplement(s) for figure 2:

**Figure supplement 1.** Phase difference between M1 and DLS 4–8 Hz LFP signals during NREM is not compatible with volume conduction.
**Figure supplement 2.** Corticostriatal NREM LFP coherence between ~5 and 11 Hz is correlated to skill learning.

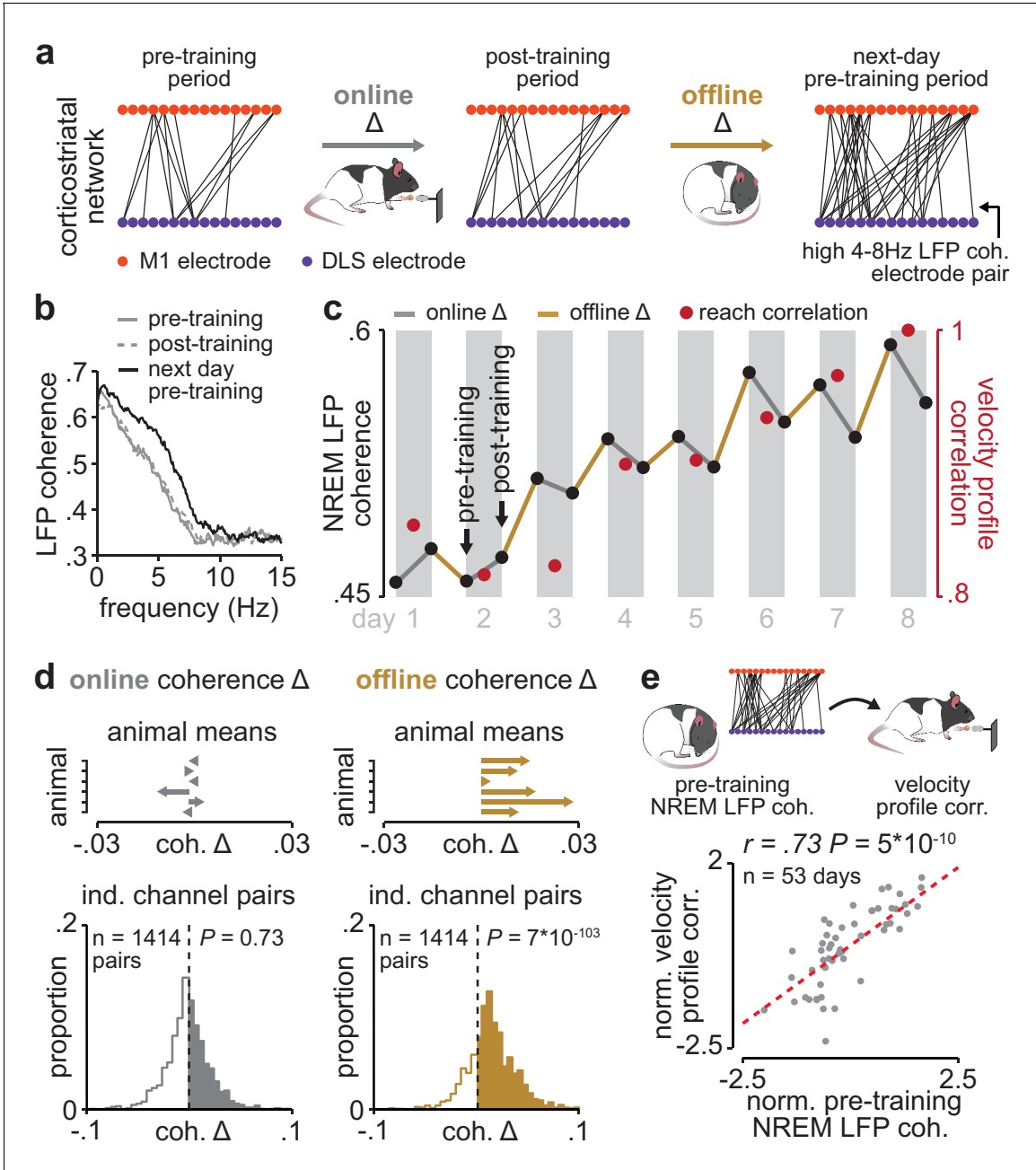

**Figure 3.** Corticostriatal functional connectivity increases offline during skill learning. (**a**) Depiction of M1 and DLS electrode pairs with high 4–8 Hz LFP coherence (>0.6 coherence value measured in NREM) during pre- and post-training periods on one day of training, and the pre-training period on the subsequent day of training, in example animal. (**b**) LFP coherence spectrums across an example M1 and DLS electrode pair measured during NREM in the pre- and post-training periods on one day of training, and the pre-training period on the subsequent day of training. (**c**) 4–8 Hz LFP coherence measured during NREM on each pre- and post-training period throughout learning for example M1 and DLS electrode pair, overlaid with reach velocity profile correlation values on each day of training. (**d**) Comparison of the mean online (left) and offline (right) change in LFP coherence (4–8 Hz, measured in NREM) across training days for all M1 and DLS electrode pairs that increased in coherence from the first to last day of training, mean in each animal (top), and histogram of all electrode pairs across animals (bottom). (**e**) Scatterplot between each day's 4–8 Hz LFP coherence measured during the pre-training period and reach velocity profile correlation value for the subsequent training period. Both values are normalized within each animal by z-scoring the values across days. M1, primary motor cortex; DLS, dorsolateral striatum; LFP, local field potential; NREM, non-rapid-eye-movement sleep.

The online version of this article includes the following figure supplement(s) for figure 3:

**Figure supplement 1.** Online and offline changes in 4–8 Hz LFP coherence in individual animals.

top three PCs) and 0.40±0.04 Pearson's *r* on last 3 training days, *P*=7×10$^{-3}$, Wilcoxon rank-sum test, *n*=11 early and late days, only days with greater than 6 M1 and DLS units recorded were considered; baseline period: 0.03±0.02 Pearson's *r* on first 3 training days and 0.02±0.02 Pearson's *r* on last 3 training days, *P*=0.28, Wilcoxon rank-sum test, *n*=11 early and late days, only days with greater than 6 M1 and DLS units recorded were considered).

It is possible that the increase in ability to predict DLS neural trajectories from M1 activity is influenced by local learning-related changes in DLS spiking activity during action execution or a change in variance explained by the top PCs of DLS activity. However, we found no significant difference in the trial-averaged spiking modulation of DLS units during action execution between early (first 3 days of training) and late (last 3 days of training) training days (*Figure 4—figure supplement 1*; 1.6±0.05 modulation value on early days and 1.7±0.09 modulation value on late days, mean± SEM, *t* (409)=−1.7, *P*=0.09, two-sample *t*-test, *n*=233 DLS units on early days and 178 DLS units on late days), as well as no significant difference in the variance explained by the first three PCs computed from DLS activity during action execution between early and late training days (58.0±2.7 variance explained on early days and 52.2±3.0 on late days, mean± SEM, *t*(27)=1.4, *P*=0.17, two-sample *t*-test, *n*=17 early days and 12 late days). A similar number of DLS units were also recorded on early and late training days (13.6±1.5 DLS units per day per animal on early days and 12.4±1.1 DLS units on late days, mean± SEM, *t*(27)=0.6, *P*=0.56, two-sample *t*-test, *n*=17 early days and 12 late days). Altogether, these results provide evidence that the increased ability to predict DLS neural trajectories during skill execution from M1 spiking activity reflects cross-area dynamics emerging with learning, rather than local learning-related changes in DLS. Consistent with the idea that offline plasticity in the corticostriatal network is relevant to skill learning, we found that 4–8 Hz LFP coherence measured during the pre-training period on each day was significantly correlated to the ability to predict DLS neural trajectories during the subsequent training period from M1 spiking activity (*Figure 4c*; reaching period: *r*=0.58, *P*=9×10$^{-4}$, Pearson's *r*, baseline period: *r*=0.03, *P*=0.87).

## Sleep spindles in NREM facilitate corticostriatal transmission

We next sought to identify neural activity patterns relevant for corticostriatal plasticity during offline periods. To do this, we first examined how corticostriatal transmission strength, that is, the degree to which M1 neural activity drives DLS activity, differed across behavioral states, as increased transmission rate may enable activity-dependent plasticity (*Charpier and Deniau, 1997*; *Figure 5a*). To measure transmission strength, we identified coupled pairs of M1 and DLS neurons based on consistent short-latency spike-timing relationships. We utilized a 'basic spike jitter' method to identify coupled pairs of M1 and DLS units with significant spike-timing relationships at timescales consistent with the conduction and synaptic delays between M1 and DLS (~6 ms time lag from M1 to DLS activity; *Koralek et al., 2013*). The spike jitter method allows for the differentiation of such short-latency spike-timing relationships from spike-timing relationships at longer time scales (>50 ms), more likely to reflect common input or slow population spiking fluctuations (*Fujisawa et al., 2008*; *Amarasingham et al., 2012*; *Hatsopoulos et al., 2003*). Across the population of recorded M1 (*n*=1100 units) and DLS neurons (*n*=579 units, 71% classified as medium spiny neurons [MSNs] based on spike width; *Figure 5—figure supplement 1*), we identified ~2.6% of pairs with a significant short-latency spiking relationship (311/12,169 pairs; *Figure 5—figure supplement 2*). It is important to note that in addition to this small percentage of neuron pairs with significant short-latency spiking relationships, we observed relationships between M1 and DLS neural activity at timescales greater than 50 ms, as can be seen in the jittered cross-correlations of M1 and DLS spiking (*Figure 5—figure supplement 2*). This is consistent with recent work that carefully dissected the relationship between cortical and striatal activity and observed broad cross-correlation histograms with peaks at a short-latency delay (~3 ms) between cortical and striatal spiking activity (*Peters et al., 2021*). The relatively broad cross-correlation histogram peaks observed between M1 and DLS neurons, compared to those typically seen between cortical neurons (*Fujisawa et al., 2008*), may be because striatal MSNs receive weak input from many cortical neurons, rather than strong input from individual cortical neurons (*Dudman and Gerfen, 2015*). Therefore, convergent activation from several cortical neurons is likely required to drive MSN spiking activity, resulting in temporal jitter that decreases the consistency of any specific M1 and DLS neuron spiking relationship.

Having characterized a population of coupled M1 and DLS neurons with consistent short-latency spike-timing relationships, we next compared corticostriatal transmission strength across sleep and

wake states by measuring the magnitude of the short-latency cross-correlation (1–15 ms time lag) within the coupled population. To account for differences in firing rate across wake and sleep states

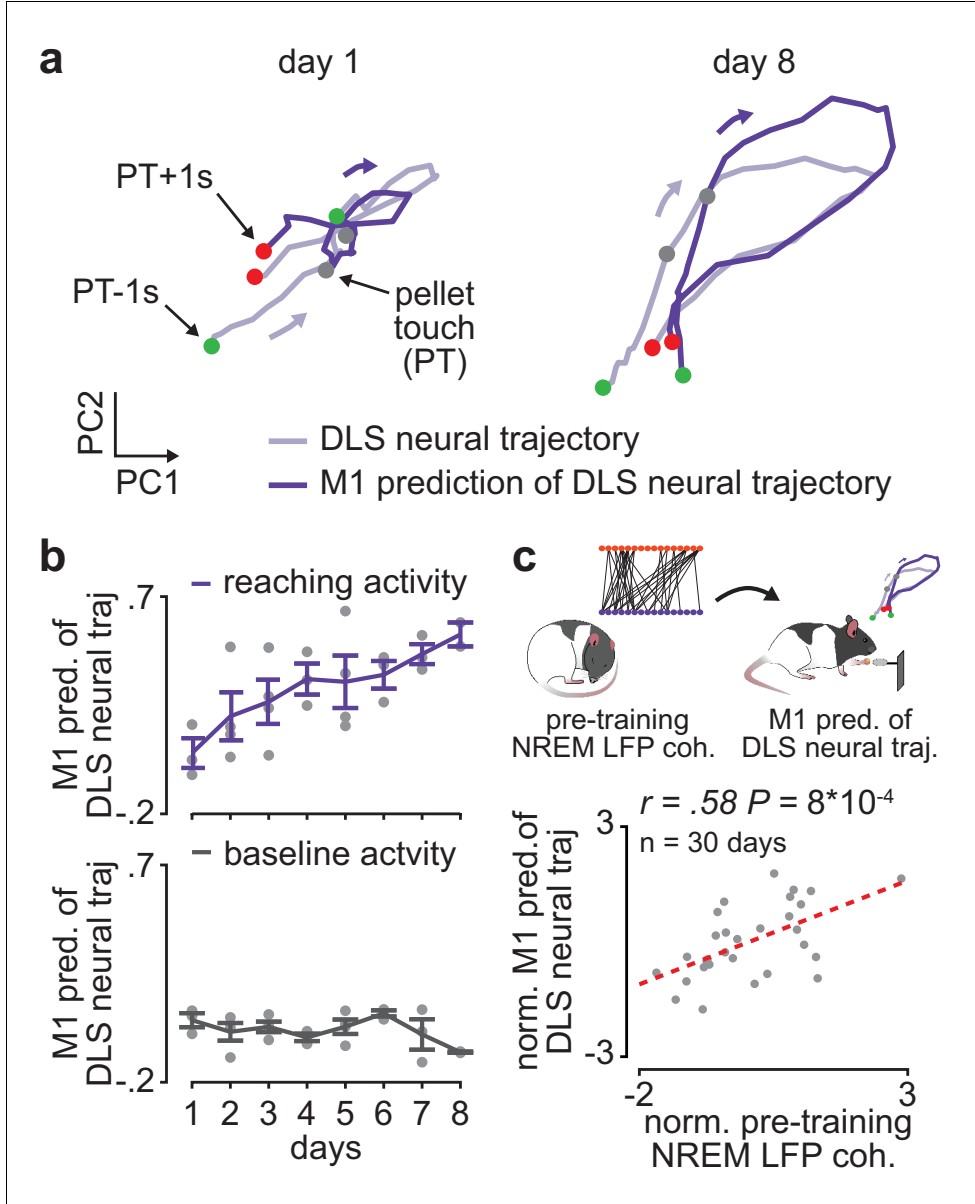

**Figure 4.** Offline increases in corticostriatal functional connectivity predict emergence of cross-area neural dynamics during subsequent skill execution. (a) Trial-averaged neural trajectory (PC1 and PC2) of DLS activity during reaching (1 s before to 1 s after pellet touch) on day 1 (left) and day 8 (right) of training in example animal, overlaid with prediction of DLS neural trajectory from M1 spiking activity. (b) Ability to predict DLS neural trajectory (PC1–3) during reaching and during a baseline, non-reaching, period from M1 spiking activity on each day of training (gray dots represent days for individual animals, mean± SEM across animals in color). (c) Correlation between each day's mean 4–8 Hz NREM LFP coherence during the pre-training period and ability to predict DLS neural trajectory (PC1–3) during reaching from M1 spiking activity during the subsequent training period. Both values are normalized within each animal by z-scoring the values across days. M1, primary motor cortex; DLS, dorsolateral striatum; LFP, local field potential; NREM, non-rapid-eye-movement sleep; PC, principal component. The online version of this article includes the following figure supplement(s) for figure 4:

**Figure supplement 1.** Comparison of corticostriatal spiking modulation during action execution on early and late training days.

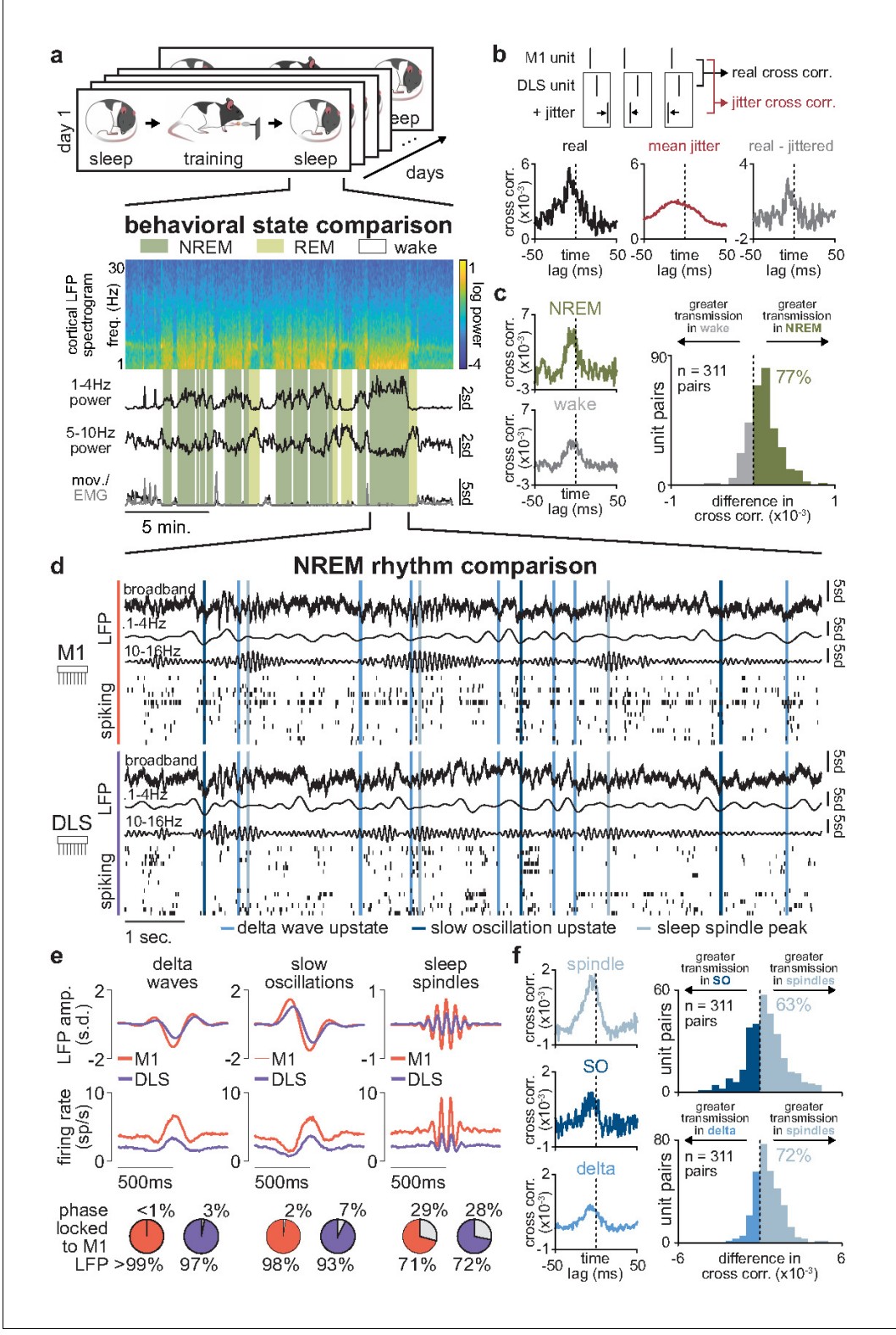

**Figure 5.** Sleep spindles in NREM facilitate corticostriatal transmission. (**a**) M1 local field potential (LFP) spectrogram and behavioral state detection from example session. (**b**) Schematic depicting basic spike jitter method for detecting coupled M1 and DLS neurons (top) and normalization by subtracting mean jittered cross-correlation from real cross-correlation (bottom). (**c**) Comparison of normalized cross-correlations of spiking activity during NREM and wake from example coupled pair of M1 and DLS units (left) and histogram of differences in

*Figure 5 continued on next page*

*Figure 5 continued*

short-latency cross-correlation magnitude (1–15 ms) between NREM and wake for all pairs of coupled M1 and DLS neurons (right). (**d**) Snippet of LFP and single-unit spiking activity from M1 and DLS during NREM overlaid with detected NREM rhythms in M1. (**e**) Mean LFP and spiking activity during slow oscillations, delta waves, and sleep spindles in both M1 and DLS in example animal (top) and percentage of M1 and DLS units across animals significantly phase locked to M1 LFP during each NREM rhythm (bottom; significance threshold of $P$=0.05, Rayleigh test of uniformity). (**f**) Comparison of normalized cross-correlations of spiking activity during NREM rhythms from example coupled pair of M1 and DLS units (left) differences in short-latency cross-correlation magnitude (1–15 ms) between NREM rhythms for all pairs of coupled M1 and DLS neurons (right). M1, primary motor cortex; DLS, dorsolateral striatum; NREM, non-rapid-eye-movement sleep.

The online version of this article includes the following figure supplement(s) for figure 5:

**Figure supplement 1.** DLS unit classification.
**Figure supplement 2.** Basic spike jitter method to identify pairs of M1 and DLS neurons with consistent short-latency spike-timing relationships.
**Figure supplement 3.** Firing rates in M1 and DLS across behavioral states.
**Figure supplement 4.** NREM rhythm detection.
**Figure supplement 5.** Corticostriatal modulation across NREM rhythms.

(*Figure 5—figure supplement 3*), we normalized each cross-correlation by subtracting the mean spike jittered cross-correlations before comparison (*Figure 5b*). We found that corticostriatal transmission was higher during NREM, compared to wake, in 77% of coupled M1 and DLS neuron pairs (*Figure 5c*), suggesting activity patterns in NREM may be particularly relevant for offline corticostriatal plasticity given the increased transmission of activity from M1 to DLS.

Given the heterogeneous nature of NREM activity, we next sought to examine whether corticostriatal transmission strength was boosted during specific patterns of activity in NREM (*Figure 5d*). We detected NREM rhythms in M1 that have been previously related to activity-dependent plasticity in cortex, including sleep spindles, SOs, and delta waves (*Ramanathan et al., 2015*; *Kim et al., 2019*; *Huber et al., 2004*; *Durkin et al., 2017*; *Figure 5—figure supplement 4*) and examined whether activity in DLS was also modulated during these rhythms. We found that both LFP signals and spiking in DLS were significantly modulated during SOs, delta waves, and sleep spindles detected in M1 (*Figure 5e*; *Figure 5—figure supplement 5*). We next compared corticostriatal transmission strength between NREM rhythms by measuring the magnitude of the short-latency cross-correlation (as above, 1–15 ms time lag, within the coupled population of M1 and DLS neurons, and normalized by subtracting the mean spike jittered cross-correlation). This revealed that corticostriatal transmission strength was greatest during sleep spindles, compared to SOs or delta waves (*Figure 5f*), suggesting that sleep spindles during NREM may be particularly relevant periods for activity-dependent plasticity within the corticostriatal network.

## Short-latency spike-timing relationships are uniquely preserved within the post-training period for sleep spindle modulated M1 and DLS neuron pairs

We further investigated the role of sleep spindles in offline corticostriatal plasticity by examining whether sleep spindle modulation impacted changes in corticostriatal transmission within pre- and post-training periods. To do this, we divided each pre- and post-training period into halves and measured the difference in short-latency cross-correlation magnitude (as above, 1–15 ms time lag) from the first to the second half of each period (*Figure 6a*). Short-latency cross-correlation values were calculated specifically within the previously identified population of coupled M1 and DLS neurons, using spiking activity during NREM to control for any differences in time spent in each behavioral state (*Figure 6—figure supplement 1*). A consistent increase or decrease in short-latency cross-correlation magnitude between halves would indicate that corticostriatal functional connectivity is modified during the pre- or post-training offline periods. To examine the role that sleep spindles may play in such offline plasticity, we also classified each M1 and DLS neuron pair based on whether both neurons were modulated by sleep spindles. This revealed that corticostriatal functional connectivity, measured by short-latency cross-correlation magnitude, was specifically preserved in spindle-modulated neuron pairs during the post-training period, in contrast to non-spindle modulated pairs during

the post-training period or all pairs during the pre-training period (*Figure 6b and c*). This change could not be attributed to differences in sleep depth, as sleep depth measured by low-frequency cortical LFP power did not show a similar trend (*Figure 6—figure supplement 2*). Altogether, this suggested that sleep spindles following training may be involved in preserving learning-related cross-area connectivity in the corticostriatal network. Strikingly, post-training period changes in short-latency cross-correlation magnitude averaged across coupled M1 and DLS neuron pairs were significantly correlated to subsequent overnight changes in 4–8 Hz LFP coherence averaged across M1 and DLS electrodes (*Figure 6d*), indicating that sleep-spindle related preservation of

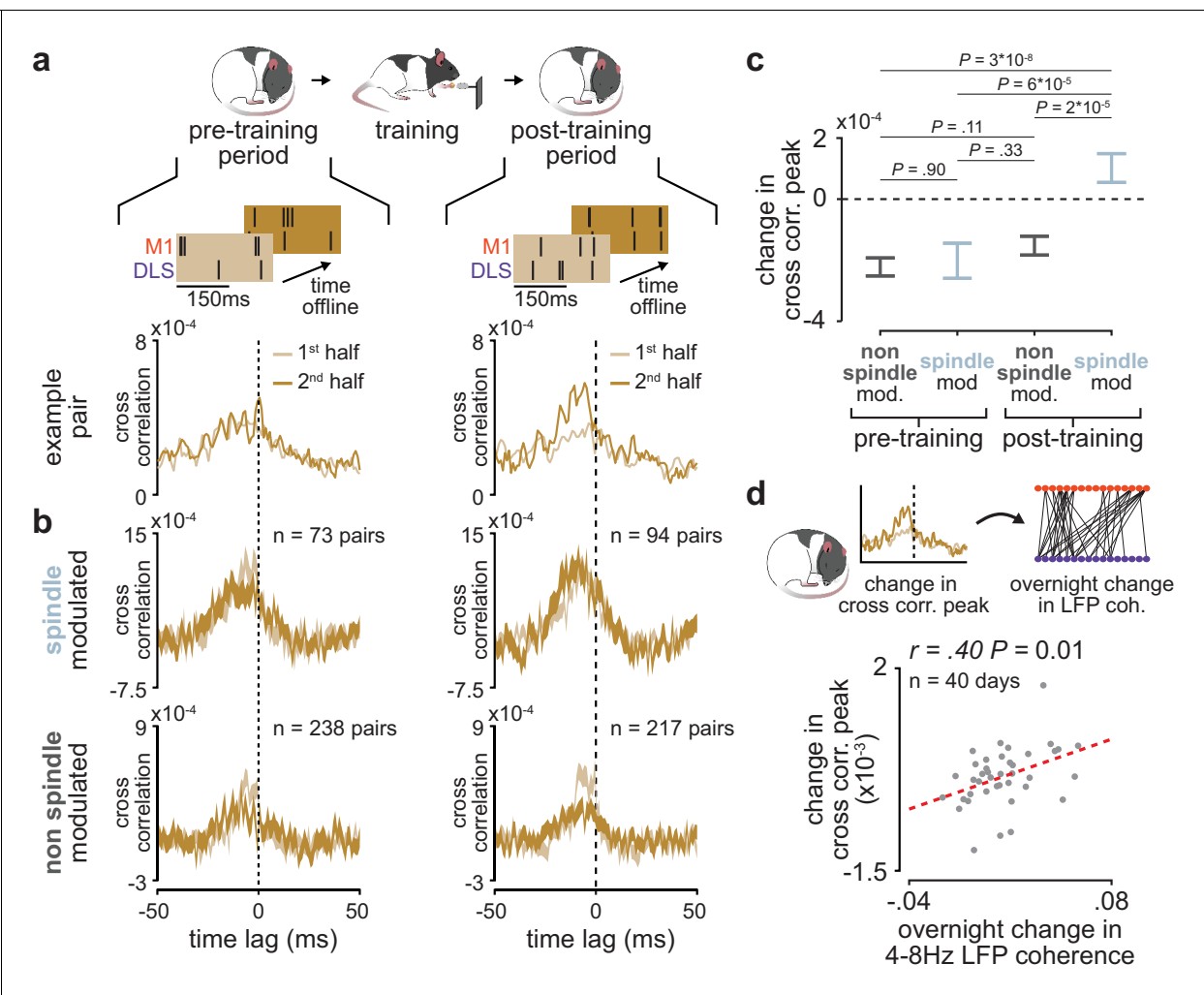

**Figure 6.** Short-latency spike-timing relationships are uniquely preserved within the post-training period for sleep spindle modulated M1 and DLS neuron pairs. (a) Schematic of changes in short-latency spike-timing relationships measured by cross-correlations of spiking activity during NREM from the first and second half of pre- (left) and post-training (right) periods for example M1 and DLS neuron pair. (b) Cross-correlations of spiking activity during NREM for coupled pairs of M1 and DLS neurons that are spindle-modulated (top) or non-spindle modulated (bottom) during the first and second half of pre- (left) and post-training (right) periods (width of line represents mean± SEM). (c) Comparison of change in short-latency cross-correlation peak (1–15 ms time lag) between spindle-modulated and non-spindle modulated M1 and DLS pairs during the pre- and post-training periods (mean± SEM). (d) Scatterplot of mean change in short-latency cross-correlation magnitude (post-training change normalized by pre-training change) across coupled M1 and DLS neurons pairs and mean overnight change in 4–8 Hz LFP coherence across M1 and DLS electrodes. M1, primary motor cortex; DLS, dorsolateral striatum; NREM, non-rapid-eye-movement sleep.

The online version of this article includes the following figure supplement(s) for figure 6:

**Figure supplement 1.** Comparison of time spent in each behavioral state during the pre- and post-training periods.

**Figure supplement 2.** Comparison of sleep depth within and across the pre- and post-training periods.

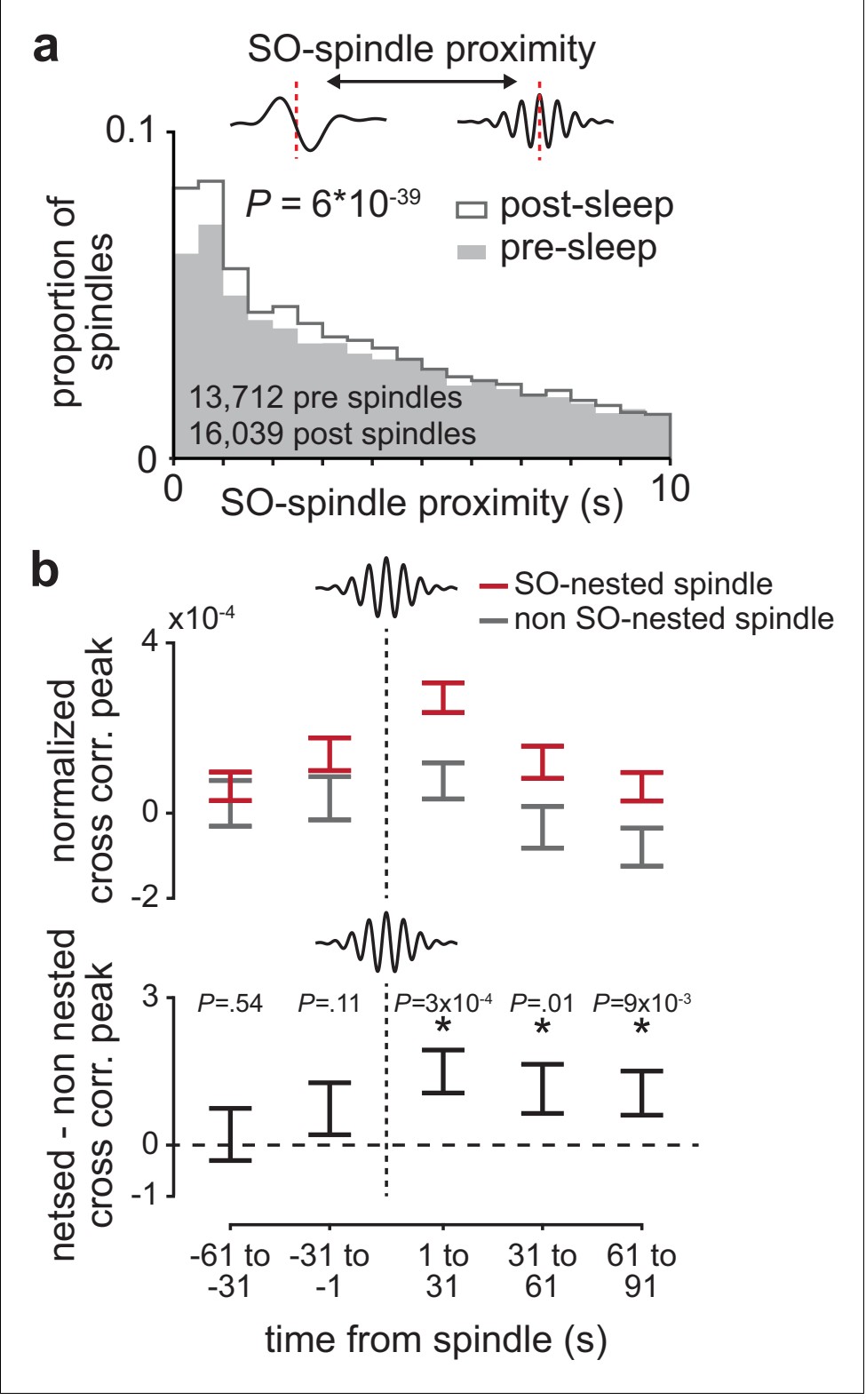

**Figure 7.** The impact of sleep spindles on corticostriatal connectivity is influenced by temporal proximity to preceding slow oscillations (SOs). (**a**) Distributions of the temporal proximity to preceding SOs for all sleep spindles during NREM in pre- and post-training periods, across days and animals. (**b**) Short-latency cross-correlation magnitude (1-15ms time lag) across coupled M1 and DLS neuron pairs calculated from spiking occurring in 30 s bins around SO-nested spindles (sleep spindles occurring within 1 s after a SO zero-crossing) and

*Figure 7 continued on next page*

*Figure 7 continued*

non-SO nested spindles (sleep spindles occurring 5 s or more after a SO zero-crossing; top), and the difference in short-latency cross-correlation magnitude between SO-nested and non-SO-nested spindles. M1, primary motor cortex; DLS, dorsolateral striatum; NREM, non-rapid-eye-movement sleep.

The online version of this article includes the following figure supplement(s) for figure 7:

**Figure supplement 1.** Spindle probability and corticostriatal firing rates around SO-nested and non-SO-nested sleep spindles.

corticostriatal functional connectivity within the first few hours after training may be related to overnight plasticity in the corticostriatal network.

## The impact of sleep spindles on corticostriatal connectivity is influenced by temporal proximity to preceding slow oscillations

Finally, we sought to understand why corticostriatal functional connectivity was preserved across spindle-modulated pairs during the post-training period, but not spindle-modulated pairs during the pre-training period. To do this, we examined the interaction between sleep spindles and SOs, a relationship known to be relevant for sleep-dependent processing (*Kim et al., 2019*; *Niethard et al., 2018*; *Silversmith et al., 2020*; *Rasch and Born, 2013*). We found a large shift in the temporal proximity to preceding SOs from the pre- to post-training period, with a larger proportion of sleep spindles in the post-training period 'nested' near SOs (*Figure 7a*; $P=6\times10^{-39}$, two-sample Kolmogorov–Smirnov test). We then examined whether SO nesting of sleep spindles influenced the role of sleep spindles in preserving corticostriatal functional connectivity. Within coupled M1 and DLS neuron pairs, we calculated the short-latency cross-correlation magnitude (as above, 1–15 ms time lag) within 30 s bins before and after every sleep spindle. We found that corticostriatal transmission strength, measured by short-latency cross-correlation magnitude, was significantly elevated after slow oscillation-nested sleep spindles (SO-nested spindles; sleep spindles within 1 s of a SO) compared to non-SO-nested sleep spindles (*Figure 7b*; non-SO-nested spindles; sleep spindles occurring at least 5 s after a SO). There were no clear differences in sleep spindle frequency or corticostriatal firing rates between SO-nested and non-SO-nested spindles that would account for this difference (*Figure 7—figure supplement 1*). This suggested that increased nesting of SOs and sleep spindles may account for the unique preservation of corticostriatal functional connectivity within spindle-modulated M1 and DLS neuron pairs during the post-training period.

## Discussion

Plasticity in cortical connectivity to the striatum can influence the balance between behavioral variability and stability (*Malvaez and Wassum, 2018*; *Lipton et al., 2019*; *Yin and Knowlton, 2006*; *Vicente et al., 2020*; *Gremel and Costa, 2013*). Here, in the context of skill learning, we provide evidence that sleep is a relevant period for such corticostriatal plasticity. We show that functional connectivity between motor cortex and striatum, measured by both LFP coherence and spike-timing relationships, evolves during offline periods away from training, rather than during training itself, and that blocking the activation of striatal NMDA receptors during these offline periods disrupts skill learning. We then identify NREM sleep spindles as uniquely poised to mediate such plasticity, through their interaction with SOs.

### NREM sleep rhythms and plasticity

Our results add to a growing body of work linking NREM rhythms to sleep-dependent plasticity (*Ramanathan et al., 2015*; *Kim et al., 2019*; *Huber et al., 2004*; *Durkin et al., 2017*; *Barakat et al., 2013*). We find that during sleep immediately following training (within ~1–3 hr), neuron pairs across M1 and DLS that are modulated during sleep spindles uniquely preserve their short-latency spike-timing relationships, interpreted as a maintenance of functional connectivity between cortex and striatum. We found that this maintenance was influenced by the temporal proximity between sleep spindles and preceding SOs, and was correlated to overnight changes in LFP coherence, suggesting that NREM rhythms during the first few hours of sleep after training may be

particularly relevant for sleep-dependent plasticity, consistent with previous work (*Miyamoto et al., 2016*).

While sleep spindles have been previously linked to plasticity (*Durkin et al., 2017*; *Barakat et al., 2013*; *Rosanova and Ulrich, 2005*; *Clawson et al., 2016*), how neural activity during sleep spindles leads to long-term plasticity remains unclear. It has been demonstrated in vitro that SO and sleep spindle activity patterns can drive NMDA receptor-dependent potentiation (*Rosanova and Ulrich, 2005*; *Chauvette et al., 2012*). Given evidence for NMDA receptor-dependent potentiation of cortical inputs to the striatum during skill learning (*Calabresi et al., 1992*; *Charpier and Deniau, 1997*), one intriguing possibility is that sleep spindles, gated by their temporal proximity to preceding SOs, promote the potentiation of cortical inputs to the striatum through NMDA receptor activation. This would be consistent with our finding that blocking striatal NMDA activation during offline periods disrupts skill learning, as well as previous work linking NMDA receptors to sleep-dependent consolidation (*Gais et al., 2008*). Importantly, however, blocking striatal NMDA receptors also impacts spontaneous striatal activity (*Pomata et al., 2008*). Further work is required to understand how striatal AP5 infusions may influence striatal activity during NREM.

We also observed a decrement in short-latency spike-timing relationships within M1 and DLS neuron pairs measured in pre-training sleep, or pairs measured in post-training sleep but not modulated during sleep spindles. This change is consistent with a growing body of work supporting the synaptic homeostasis hypothesis (SHY), which proposes that sleep drives the general homeostatic downscaling of synapses which are upregulated during wake (*Tononi and Cirelli, 2014*). Such general downscaling can support memory consolidation indirectly by increasing the signal-to-noise of memory representations encoded during wake (*Rasch and Born, 2013*; *Miyamoto et al., 2021*). An outstanding question is how the general downscaling of synapses during sleep proposed in SHY may interact with the preservation or potentiation of specific synapses relevant to learning (*Rasch and Born, 2013*; *Miyamoto et al., 2021*). Recent work suggests a way to reconcile both ideas, demonstrating that activity during sleep may preserve activity patterns generated during learning while also downscaling task-irrelevant activity (*Kim et al., 2019*; *Gulati et al., 2017*). As NREM sleep rhythms have been linked to both processes (*Kim et al., 2019*; *Huber et al., 2004*; *Gulati et al., 2017*; *Norimoto et al., 2018*), it will be important to explore how NREM rhythms differentially impact downscaling versus preserving/strengthening synapses for nearby neurons in the same brain region or connected neurons across different brain regions.

It is important to note that in this work, we measure only functional measures of corticostriatal connectivity, including LFP coherence and spike-timing relationships across M1 and DLS. One possibility is that these functional measures of connectivity reflect changes in the synaptic strength of M1 projections to the DLS. This would be consistent with evidence for the strengthening of cortical inputs to the striatum with motor training (*O'Hare et al., 2016*; *Rothwell et al., 2015*; *Yin et al., 2009*). An alternative possibility is that coordinated inputs to both M1 and DLS drive increased functional connectivity. We believe our results are most consistent with a physical change in synaptic strength, as we measured increased functional connectivity during both NREM, reflected as increased LFP coherence, as well as during awake task performance, reflected in the emergence of coupled cross-area dynamics. However, future work is required to determine whether our observations are consistent with sleep-related structural changes in synaptic strength.

## Skill learning in the corticostriatal network

Here, we show that 4–8 Hz LFP coherence across M1 and DLS measured during NREM closely tracks the emergence of a stable skilled reaching behavior, as measured by the emergence of a stable day-to-day reaching velocity profile. This is consistent with previous work showing increased coordination of M1 and DLS neural activity with skill acquisition (*Santos et al., 2015*; *Lemke et al., 2019*; *Koralek et al., 2013*), suggesting that increased communication and connectivity between cortex and striatum may be a central feature of stable skilled behavior. We also found that the emergence of a stable day-to-day reaching velocity profile was correlated to peak single-trial reaching velocity, consistent with the idea that movement velocity is a relevant aspect of skill learning (*Lemke et al., 2019*; *Hikosaka et al., 2013*). Intriguingly, here, we find that functional connectivity between M1 and DLS increases offline, rather than during training itself. This is consistent with a range of studies demonstrating that sleep benefits speed and consistency in motor tasks in humans (*Fischer et al., 2002*; *Walker et al., 2002*) and rodents (*Ramanathan et al., 2015*; *Nagai et al., 2017*), as well as

rodent brain-machine interface (BMI) tasks (*Gulati et al., 2014*; *Kim et al., 2019*). As the basal ganglia are an important regulator of movement vigor (*Dudman and Krakauer, 2016*), future work is required to determine how increases in coupling between motor cortex and striatum precisely relate to changes in the consistency and vigor of movement.

While there is growing evidence that neural signals across cortex and striatum grow more coordinated during skill learning (*Santos et al., 2015*; *Lemke et al., 2019*; *Koralek et al., 2013*; *Costa et al., 2004*), the relative importance of the 'direction' of communication between cortex and the striatum remains unclear. On one hand, it is well-established that cortical activity influences striatal activity (*Peters et al., 2021*), and that, in turn, the basal ganglia is connected to brain stem regions that control movement (*McElvain et al., 2021*). On the other hand, there is evidence that DLS activity may be important for stabilizing cortical activity patterns (*Koralek et al., 2012*; *Lemke et al., 2019*), suggesting a role for basal ganglia 'feedback' to cortex through the thalamus (*Aoki et al., 2019*; *Athalye et al., 2020*). Recent work demonstrated the importance of thalamic input for reliable cortical neural dynamics (*Sauerbrei et al., 2020*). One intriguing possibility is that the nature of corticostriatal communication evolves during learning. For example, cortical input to striatum may be essential during the initial acquisition of a skilled movement, while striatal feedback to cortex becomes important for well-learned stable and skilled movements (*Lemke, 2020*). Future work is required to determine whether sleep may facilitate changes in the direction of communication between cortex and striatum.

In summary, our results suggest a role for sleep in modifying cross-area connectivity across cortex and striatum that, in turn, impacts behavioral stability and network activity during skill learning. One important extension of this work is to explore whether sleep can impact corticostriatal connectivity in the context of maladaptive behavioral stability, such as addiction, that has been linked to the corticostriatal network (*Lipton et al., 2019*; *Gerdeman et al., 2003*). Recent work suggests that modulating NREM rhythms can regulate memory consolidation versus forgetting (*Kim et al., 2019*). It will be informative to determine whether similar manipulations could be used in the context of maladaptive stability to provide a therapeutic benefit.

# Materials and methods

## Animal care and surgery

This study was performed in strict accordance with guidelines from the USDA Animal Welfare Act and United States Public Health Science Policy. Procedures were in accordance with protocols approved by the Institutional Animal Care and Use Committee at the San Francisco Veterans Affairs Medical Center. Experiments were conducted with 12 male Long-Evans rats (approximately 12–16 weeks old) housed under controlled temperature and a 12 hr light/12 hr dark cycle with lights on at 6:00 a.m. All behavioral experiments were performed during the light period. Surgical procedures were performed using sterile techniques under 2–4% isoflurane. Six animals were implanted with either microwire electrodes (*n*=five animals; 32 or 64 channel 33 μm diameter Tungsten microwire arrays with ZIF-clip adapter; Tucker-Davis Technology) or high-density silicon probes (*n*=one animal; 256 channel custom-built silicon probes; *Egert et al., 2020*) targeted to the forelimb area of M1 (centered at 3.5 mm lateral and 0.5 mm anterior to bregma and implanted in layer V at a depth of 1.5 mm) and the DLS (centered at 4 mm lateral and 0.5 mm anterior to bregma and implanted at a depth of 4 mm). Six additional animals were implanted with infusion cannulas (PlasticsOne; 26 Ga) targeted to the DLS. Surgery involved exposure and cleaning of the skull, preparation of the skull surface (using cyanoacrylate), and implantation of skull screws for overall headstage stability. In the animals implanted with neural probes, a reference screw was implanted posterior to lambda, contralateral to the neural recordings and a ground screw was implanted posterior to lambda, ipsilateral to the neural recordings. Craniotomy and durectomy were then performed, followed by implantation of neural probes or infusion cannulas and securing of the implant with Kwik-Sil (World Precision Instruments), C and B Metabond (Parkell, Product #S380), and Duralay dental acrylic (Darby, Product #8830630). Final location of electrodes was confirmed by electrolytic lesion. In two of the animals implanted with neural probes, the forearm was also implanted with a pair of twisted EMG wires (0.007 in. single-stranded, Teflon-coated, stainless steel wire; A-M Systems) with a hardened epoxy ball (J-B Weld Company) at one end preceded by 1–2 mm of uncoated wire under the ball. Wires

were inserted into the muscle belly and pulled through until the ball came to rest on the belly. EMG wires were braided, tunneled under the skin to a scalp incision, and soldered into an electrode interface board (ZCA-EIB32, Tucker-Davis Technology). The postoperative recovery regimen included administration of buprenorphine at 0.02 mg/kg, meloxicam at 0.2 mg/kg, dexamethasone at 0.5 mg/kg, and trimethoprim/sulfadiazine at 15 mg/kg, administered postoperatively for 5 days. All animals recovered for at least 1 week before the start of behavioral training.

## In vivo electrophysiology

Spiking activity, LFP, and EMG activity were recorded using an RZ2 system (Tucker-Davis Technologies). For neural activity recorded with microwire electrode arrays, spiking data was sampled at 24,414 Hz and LFP/EMG data was sampled at 1017 Hz. To detect spikes in microwire-implanted animals, an online threshold was set using a standard deviation of 4.5 (calculated over a 5 min baseline period). Waveforms and timestamps were stored for any event that crossed below that threshold. Spike sorting was performed using Offline Sorter v.4.3.0 (Plexon) with a PCA-based clustering method followed by manual inspection. Spikes were sorted separately for each day, combining the pre-training, training, and post-training periods. Units were accepted based on waveform shape, clear cluster boundaries in PC space, and 99.5% of detected events with an ISI>2 ms. Neural activity recorded with silicon probes was recorded at 24,414 Hz. Spike times and waveforms were detected from the broadband signal using Offline Sorter v.4.3.0 (Plexon). Spike waveforms were then sorted using Kilosort2 (https://github.com/MouseLand/Kilosort2; *Pachitariu, 2020*). We accepted units based on manual inspection using Phy (https://github.com/cortex-lab/phy; *Buccino et al., 2021*) and 99.5% of detected events with an ISI>2 ms.

## Viral injection (*Figure 1*)

To label anterograde projections in M1, we injected 750 nl of AAV8-hsyn-JAWs-KGC-GFP-ER2 virus into two sites (1.5 mm anterior, 2.7 mm lateral to bregma, at a depth of 1.4 mm and 0.5 mm posterior, 3.5 mm lateral to bregma, at a depth of 1.4 mm). Two weeks after injection rats were anesthetized and transcardially perfused with 0.9% sodium chloride, followed by 4% formalin. The harvested brains were post-fixed for 24 hr and immersed in 20% sucrose for 2 days. Coronal cryostat sections (40 µm thickness) were then mounted and imaged with a fluorescent microscope.

## Reach-to-grasp task (*Figures 1*, *3* and *4*)

Rats naïve to any motor training were first tested for forelimb preference. This involved presenting approximately 10 food pellets to the animal and observing which forelimb was most often used to reach for the pellet. Rats then underwent surgery for either neural probe or cannula implantation in the hemisphere contralateral to the preferred hand. Following the recovery period, rats were trained on the reach-to-grasp task using an automated reach-box, controlled by custom MATLAB scripts and an Arduino microcontroller. This setup requires minimal user intervention, as described previously (*Wong et al., 2015*). Each trial consisted of a pellet dispensed on the pellet tray followed by an alerting beep indicating that the trial was beginning, then the door would open. Animals had 15 s to reach, grasp, and retrieve the pellet or the trial would automatically end, and the door would close. A real-time 'pellet detector' using an infrared sensor centered over the pellet would determine when the pellet was moved, indicating the trial was over and, after 2 s, the door would close. Trials were separated by a 10-s inter-trial interval. All trials were captured by a camera placed on the side of the behavioral box (n=2 animals monitored with a Microsoft LifeCam at 30 frames/s; n=10 animals monitored with a Basler ace acA640-750uc at 75 frames/s). For animals implanted with neural probes, each animal underwent 5–14 days of training (~100–150 trials per day). For the infusion cannula implanted animals, each animal underwent 10 days of training (100 trials per day). Reach trajectories were captured from video using DeepLabCut (*Mathis et al., 2018*) to track the center of the rat's hand as well as the food pellet. We specifically analyzed reach trajectories from 500 ms before to 500 ms after 'pellet touch,' which was classified as the frame in which the hand was closest to the pellet, before the pellet was displaced off the pellet holder. Only trials in which the pellet was displaced off the pellet holder were considered. We assessed behavioral consistency throughout training in both neural probe and cannula implanted animals by calculating the correlation between the mean velocity profile of reaches on each day of training and the mean velocity profile of reaches

on the last day of training which served as the learned 'template.' These correlations were computed separately for the x and y dimensions and then averaged. To calculate total velocity profile correlation change, the last day, which served as template, was excluded in both neural probe and cannula implanted animals. We also generated shuffled distributions to test the significance of the effect of AP5 infusions (compared to saline) on velocity profile correlation and single-trial peak reaching velocity. To do this, we first computed the day-to-day changes in either measure (for velocity profile correlation we excluded the last day which served as 'template'). We then computed the real effect of AP5 infusions (compared to saline) by taking the difference between the mean day-to-day change with either post-training AP5 or saline infusion, across animals. We then randomly reshuffled the AP5/saline labels and recomputed the difference 10,000 times. To generate a P value, we measured the percentile of the difference from the real data within the shuffled distribution of differences.

### DLS infusions (*Figure 1*)

To test if the offline activation of striatal NMDA receptors is required for skill learning, we infused 1 μl of either saline or NMDA blocker AP5 (5 μg/μl) at an infusion rate of 200 nl/min into the DLS immediately following training in six animals for 10 consecutive days. During the first 5 days of training, we infused three rats with AP5 and three rats with saline. During the second 5 days, we switched the infusions, that is, animals that received AP5 in the first 5 days, received saline for the second 5 days, and vice-versa.

### Neural data analyses (*Figures 2–7*)

All neural data analyses were conducted using MATLAB 2019a (MathWorks) and functions from the EEGLAB (http://sccn.ucsd.edu/eeglab/) and Chronux (http://chronux.org/) toolboxes.

### Offline behavioral state classification (*Figures 2–7*)

During each training day, neural signals were monitored during a 2–3 hr pre- and post-training period. A video was also captured from a camera placed above the behavioral box (Microsoft Life-Cam at one frame/s). Behavioral states (wake and sleep states) were classified using cortical LFP signals and movement, measured either by video or EMG activity if animals were implanted with an EMG wire. LFP was preprocessed by artifact rejection, including manual rejection of noisy electrodes and z-scoring of each electrode's signal across the entire recording session. A mean LFP signal was then generated in M1 for sleep classification by averaging across all M1 electrodes. This mean M1 LFP signal was then segmented into non-overlapping 10 s windows. In each window, the power spectral density was computed using the Chronux function *mtspecgramc*. Delta power (1–4 Hz) and theta ratio (5–10 Hz/2–15 Hz) were computed and used for behavioral state classification. Within each pre- and post-training period, mean values of delta power and theta ratio were then computed and used as thresholds for behavioral state classification: epochs with high delta power (greater than mean delta) and no movement were classified as NREM, epochs with high theta ration (greater than mean theta) and low delta power (less than mean delta) were classified as REM, and all other epochs were classified as wake. All consecutive NREM or REM epochs that were less than 60 s long (six consecutive epochs) were reclassified as wake.

### Measuring corticostriatal functional connectivity using LFP coherence (*Figures 2–4* and *6*)

To examine changes in corticostriatal functional connectivity across days, we measured LFP coherence during NREM across all M1 and DLS electrode pairs on each pre- and post-training period. On each day, we first applied common-mode referencing on M1 and DLS LFP signals using the median signal in each region, that is, at every time-point, the median signal across all electrodes in a region was calculated and subtracted from every electrode in that region to decrease common noise and minimize volume conduction. LFP coherence was then computed for LFP signals during NREM in nonoverlapping 10 s windows using chronux function *cohgramc*. For each pre- and post-training period, we classified 'high coherence LFP pairs' as pairs of M1 and DLS electrodes with a mean 4–8 Hz coherence during NREM>0.6. When comparing LFP coherence changes occurring online (from the pre- to post-training period on the same day) and offline (from the post-training period on 1 day to the pre-training period on the next day), we computed a single online and offline change value

per M1 and DLS electrode pair by averaging online and offline change across training days. To determine the relationship between 4–8 Hz LFP coherence and behavior, we averaged LFP coherence across electrodes in the pre-training session and compared that value to behavior during the subsequent training period on that day. To determine the relationship between LFP coherence and velocity profile correlation values accounting for single-trial peak velocity, we computed a Pearson linear partial correlation coefficient using MATLAB function *partialcorr*.

### Measuring the phase difference between M1 and DLS 4–8 Hz LFP signals (*Figure 2*)

To calculate the mean phase difference between M1 and DLS 4– and 8 Hz LFP signals in NREM, we filtered M1 and DLS LFP signals during NREM in each pre- and post-training period using the EEGLAB function *eegfilt.* We then extracted the phase of the filtered LFP signals using the MATLAB function *hilbert* and computed the difference between M1 and DLS signals.

### Measuring M1 LFP–DLS spike phase locking (*Figure 2*)

To compare M1-DLS 4–8 Hz LFP coherence to a distinct measure of corticostriatal functional connectivity, we calculated the phase locking of DLS units to 4–8 Hz M1 LFP in NREM during each pre- and post-training period. To measure phase locking, we filtered M1 LFP signals during NREM between 4 and 8 Hz using the EEGLAB function *eegfilt* and extracted the phase of the filtered LFP signals using the MATLAB function *hilbert*. Then, for each DLS unit simultaneously recorded, we then computed a histogram of the M1 phase at each spike time. We then computed the circular standard deviation (cSD) of these histograms using Matlab toolbox *circstats* (https://www.mathworks.com/matlabcentral/fileexchange/10676-circular-statistics-toolbox-directional-statistics). We used this cSD value as our measure of phase locking, with low cSD representing a 'peakier' histogram and therefore greater phase locking. We then compared the mean cSD for each M1 electrode (averaged across DLS units simultaneously recorded) to the mean M1-DLS 4–8 Hz LFP coherence for that M1 electrode (averaged across DLS electrodes).

### Measuring corticostriatal network dynamics during action execution (*Figure 4*)

To measure corticostriatal network dynamics during action execution, we extracted low-dimensional representations of DLS activity by performing PCA using MATLAB function *pca*. For each DLS unit, spiking activity during each trial was binned at 100 ms from 5 s before to 5 s after pellet touch and then concatenated across trials. PCA was computed on a matrix of DLS units by time bins (number of trials * 100 bins per trial). DLS activity from 1 s before to 1 s after pellet touch on each trial was then projected onto the first three PCs to generate low-dimensional neural trajectory representations of population spiking activity in DLS during action execution. We then fit a linear regression model to predict DLS neural trajectories from single-unit spiking activity in M1. A separate model was used to predict activity projected onto each of the first three PCs, using MATLAB function *fitlm* and fivefold cross-validation. For each time bin of the DLS neural trajectory, the preceding 1.5 s of spiking activity for all M1 units, binned at 100 ms, were used as predictors. The ability to predict DLS activity from M1 activity on each day was measured by averaging the correlation values from correlating the actual DLS neural trajectories and the predicted trajectories. The same method was also used to predict a baseline, non-reaching, period from 5 s to 4 s before pellet touch.

### Measuring spiking modulation during action execution (*Figure 4*)

To measure spiking modulation during action execution, spiking activity during each trial was binned at 25 ms from 5 s before to 5 s after pellet touch. Spiking activity was then averaged across trials and *z*-scored (separately for each M1 and DLS unit on each training day). Spiking modulation was then calculated by taking the sum of the absolute value of the *z*-scored activity from 1 s before to 500 ms after pellet touch divided by the sum of the absolute value of the *z*-scored activity from 3.5 s before pellet touch to 2 s before pellet touch.

## Identifying coupled M1 and DLS neuron pairs (*Figures 5–7*)

We used a 'basic spike jitter' method to identify pairs of M1 and DLS neurons with consistent short-latency spike-timing relationships (*Hatsopoulos et al., 2003*; *Fujisawa et al., 2008*; *Amarasingham et al., 2012*). Briefly, we binned at 1 ms and concatenated together the spiking activity during the first 5 min of NREM of both the pre- and post-training period (10 min total) for each pair of M1 and DLS units on each day of training. We then calculated the mean value of the short-latency cross-correlation for each pair (1–15 ms time lag centered on DLS spiking, such that positive time lags corresponded to DLS spiking after M1 spiking; consistent with the conduction and synaptic delay between M1 and DLS; *Koralek et al., 2013*). We then generated a 'jittered' distribution of short-latency cross-correlation values by jittering each DLS spike within a 50 ms window centered on the spike and recalculating the cross-correlation, repeated 1000 times. To perform the jittering, a 50 ms window is centered on each DLS spike, and the spike is replaced by one randomly chosen within that window. This method destroys any consistent spike-timing relationship at timescales smaller than the jitter window, while preserving spiking relationships on timescales greater than the jitter window. We classified a pair of M1 and DLS units as 'coupled' if the real mean short-latency cross-correlation value was greater than the 99th percentile of the jittered distribution.

## Measuring corticostriatal transmission strength (*Figure 5*)

To compare corticostriatal transmission strength between NREM and wake, we calculated cross-correlations of spiking activity binned at 1 ms from each behavioral state (NREM and wake, pre- and post-training periods concatenated together) for all coupled M1 and DLS neuron pairs ('coupling' was based on the jittering method described above). To account for firing rate differences across behavioral states, we normalized each M1 and DLS pair's cross-correlation in each behavioral state by subtracting a mean jitter cross-correlation for that behavioral state generated by repeating the jittering processes described above 1000 times and taking the average of the 1000 jittered cross-correlations. The mean short-latency cross-correlation magnitude (1–15 ms time lag centered on DLS spiking, such that positive time lags corresponded to DLS spiking after M1 spiking) was then compared between NREM and wake (rats did not spend enough time in REM sleep to make a robust comparison). To compare corticostriatal transmission strength across NREM rhythms, we calculated cross-correlations of spiking activity binned at 1 ms from each NREM rhythm (sleep spindles, delta waves, and SOs, rhythms during pre- and post-training periods were concatenated together) for all coupled M1 and DLS neuron pairs. For sleep spindles, 1 s of spiking centered on sleep spindle peak (−500 ms to 500 ms) was included from each spindle. For SOs and delta waves, a 1-s window around upstate peak (−500 ms to 500 ms) was used. The same normalization was applied as in comparisons across behavioral states (subtraction of mean jittered cross-correlation from the real cross-correlation). The mean short-latency cross-correlation magnitude (1–15 ms time lag) was then compared between NREM rhythms.

## NREM rhythm detection (*Figure 5*)

The NREM rhythm detection applied here is based on a previously used detection algorithm (*Kim et al., 2019*; *Silversmith et al., 2020*). Briefly, a mean LFP signal was generated in M1 by averaging across all electrodes. To detect sleep spindles, this mean signal was filtered in the spindle band (10–16 Hz) using a zero-phase shifted, third-order Butterworth filter. A smoothed envelope was calculated by computing the magnitude of the Hilbert transform of this signal then convolving it with a Gaussian window. Next, we determined two upper thresholds for spindle detection based on the mean and standard deviation (s.d.) of the spindle band envelope during NREM. Epochs in which the spindle envelope exceeded 2.5 s.d. above the mean for at least one sample and the spindle power exceeded 1.5 s.d. above the mean for at least 500 ms were detected as spindles. Then, spindles that were sufficiently close in time (<300 ms) were combined. To detect SOs and delta waves, the mean M1 signal was filtered in a low-frequency band (second order, zero phase shifted, high pass Butterworth filter with a cutoff at 0.1 Hz followed by a fifth order, zero phase shifted, low pass Butterworth filter with a cutoff at 4 Hz). Next, all positive-to-negative zero crossings during NREM were identified, along with the previous peaks, the following troughs, and the surrounding negative-to-positive zero crossings. Each identified epoch was considered a SO if the peak was in the top 15% of peaks, the trough was in the top 40% of troughs and the time between the negative-to-

positive zero crossings was greater than 300 ms but did not exceed 1 s. Each identified epoch was considered a delta wave if the peak was in the bottom 85% of peaks, the trough was in the top 40% of troughs and the time between the negative-to-positive zero crossings was greater than 250 ms.

### NREM rhythm modulation (*Figure 5*)

To measure the sleep spindle modulation of individual M1 and DLS units, spiking during each sleep spindle was time locked to the peak of the filtered LFP and binned at 10 ms. Spiking was averaged across sleep spindles and modulation was calculated by taking the minimum to maximal firing rate bin in the second around sleep spindle peak (−500 ms to 500 ms) divided by the minimum to maximal firing rate bin in a second long-baseline period before each spindle (−1500 ms to −500 ms relative to spindle peak). To determine SO and delta wave modulation of individual M1 and DLS units, spiking during each SO or delta wave was time locked to the peak of the upstate and binned at 10 ms. Spiking was averaged across SOs or delta waves and modulation was calculated by taking the minimum to maximal firing rate bin in the second around upstate peak (−500 ms to 500 ms) divided by the minimum to maximal firing rate bin in a second long-baseline period before each SO or delta wave (−1500 ms to −500 ms relative to upstate peak).

### Measuring changes in corticostriatal transmission strength within pre- and post-training periods (*Figure 6*)

To measure changes in corticostriatal transmission strength within pre- and post-training periods, we calculated cross-correlations of spiking activity binned at 1 ms from NREM activity during the first and second half of each pre- and post-training period for all coupled M1 and DLS neuron pairs. We compared changes in the mean short-latency cross-correlation magnitude (1–15 ms) from the first to second half of each pre- and post-training period between coupled M1 and DLS pairs that were spindle modulated and non-spindle modulated. To determine spindle modulated pairs, we generated peri-event time histograms (PETHs) of spiking activity for each M1 and DLS unit, locked to sleep spindle peak and binned in 10 ms bins from 2 s before to 2 s after spindle peak (400 bins), averaged across all spindles. Sleep spindle modulation was then calculated by taking the minimum to maximal firing rate bin within the 1 s period centered on spindle peak (−500 ms to 500 ms). We then generated a distribution of shuffled modulations by shuffling the time bins and recalculating the modulation of this shuffled PETH. This shuffling procedure was repeated 1000 times to generate a distribution. Units with a non-shuffled modulation greater than the 99% percentile of the shuffled distribution were considered significantly sleep spindle modulated. Spindle modulated pairs included both a spindle modulated M1 and DLS unit and all other pairs were considered non-spindle modulated.

### Measuring changes in sleep depth within pre- and post-training periods (*Figure 6*)

To measure changes in sleep depth within pre- and post-training periods, we first generated a mean LFP signal in each period by averaging across all M1 electrodes. This mean M1 LFP signal was then separated into the first and second half of each pre- and post-training period and segmented into non-overlapping 10 s window. Power spectral density was then computed in each window using the Chronux function *mtspecgramc* and then averaged over low frequencies (1–4 Hz) as a proxy for sleep depth. We interpolated the low-frequency power values in each pre- and post-training period to normalize duration across days and animals.

### Determining temporal proximity between slow oscillations and sleep spindles (*Figure 7*)

SO to sleep spindle proximity was determined by measuring the temporal proximity between each sleep spindle peak and the preceding SO zero-crossing (positive to negative LFP). SO-nested spindles were defined as spindles occurring with 1 s of a SO zero-crossing and non-SO-nested spindles were defined as spindles occurring at least 5 s after a SO zero-crossing.

## Measuring corticostriatal transmission strength dynamics around sleep spindles (*Figure 7*)

To determine corticostriatal transmission strength changes occurring around sleep spindles, we calculated cross-correlations of spiking activity binned at 1 ms in 30 s bins around every sleep spindle. These bins were placed $-91$ s to $-61$ s, $-61$ s to $-31$ s, $-31$ s to $-1$ s, 1 s to 30 s, 31 s to 61 s, and 61 s to 91 s around a spindle as to avoid including spiking activity during the spindle itself. Corticostriatal transmission strength was calculated in each bin by averaging the short-latency cross-correlation values for each coupled M1 and DLS pair in each bin (1–15 ms time lag centered on DLS spiking, such that positive time lags corresponded to DLS spiking after M1 spiking). Corticostriatal transmission strength values around each spindle (six total bins) were then normalized by subtracting by the value of the first bin ($-91$ s to $-61$ s). Changes in corticostriatal transmission strength were then compared between SO-nested and non-SO-nested spindles. The same normalization was used to compute changes in M1 and DLS firing rates, as well as sleep spindle probability, around each sleep spindle.

## Additional information

### Funding

| Funder | Grant reference number | Author |
|---|---|---|
| Veterans Health Administration HSR and D | I01RX001640-06 | Karunesh Ganguly |
| National Institute of Mental Health | R01MH111871-04 | Karunesh Ganguly |
| Horizon 2020 - Research and Innovation Framework Programme | 895379 | Stefan M Lemke |

The funders had no role in study design, data collection and interpretation, or the decision to submit the work for publication.

### Author contributions

Stefan M Lemke, Conceptualization, Data curation, Software, Formal analysis, Validation, Investigation, Visualization, Writing - original draft, Project administration, Writing - review and editing; Dhakshin S Ramanathan, Conceptualization, Data curation, Formal analysis, Writing - review and editing; David Darevksy, Conceptualization, Data curation, Formal analysis; Daniel Egert, Data curation, Validation, Methodology; Joshua D Berke, Methodology; Karunesh Ganguly, Conceptualization, Supervision, Writing - original draft, Project administration, Writing - review and editing

### Author ORCIDs

Stefan M Lemke ![iD] https://orcid.org/0000-0002-1721-5425
Joshua D Berke ![iD] http://orcid.org/0000-0003-1436-6823
Karunesh Ganguly ![iD] https://orcid.org/0000-0002-2570-9943

### Ethics

Animal experimentation: This study was performed in strict accordance with guidelines from the USDA Animal Welfare Act and United States Public Health Science Policy. Procedures were in accordance with protocols approved by the Institutional Animal Care and Use Committee at the San Francisco Veterans Affairs Medical Center (Protocol 19-002).

### Decision letter and Author response

Decision letter https://doi.org/10.7554/eLife.64303.sa1
Author response https://doi.org/10.7554/eLife.64303.sa2

## Additional files

### Supplementary files

• Transparent reporting form

### Data availability

The data and corresponding code used for analyses is available on Dryad.

The following dataset was generated:

| Author(s) | Year | Dataset title | Dataset URL | Database and Identifier |
|---|---|---|---|---|
| Lemke SM, Ramanathan DS, Darevksy D, Egert D, Berke JD, Ganguly K | 2021 | Data from: Coupling between motor cortex and striatum increases during sleep over long-term skill learning | https://doi.org/10.7272/Q6KK9927 | Dryad Digital Repository, 10.7272/Q6KK9927 |

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
