## [Decision Letter]

**Acceptance summary:**

This work is a thought-provoking study of the interaction between sleep and corticostriatal plasticity. The reviewers agreed that there are many strengths of the study and it could spark new directions of research.

**Decision letter after peer review:**

Thank you for sending your article entitled "Sleep spindles coordinate corticostriatal reactivations during the emergence of automaticity" for peer review at *eLife*. Your article is being evaluated by 3 peer reviewers, and the evaluation is being overseen by a Reviewing Editor and Michael Frank as the Senior Editor.

Essential revisions:

The reviewers felt that the topic of the paper is very interesting and the study is original and creative. However, a number of issues were raised about clarity of the presentation, data analysis, and interpretation of the results. The concerns fall under the following categories:

1. Trajectory consistency and movement speed

Throughout the manuscript, the authors define automaticity has an increased consistency in reaching trajectory (abstract, line 23-26, line 86) but the main measurement used in figure 1 is the correlation of the velocity profile (which actually shows a strong increase in speed during learning). This is problematic for several reasons:

– An inattentive reader may think that such that the y-axis label "reach correlation" in panels e, f, and g of figure 1 is referring to the correlations of the trajectories shown in c.

– This choice raises the question of why the authors quantified behavior by looking at the consistency of the speed profile rather than directly through trajectories correlation. Looking at the traces of the trajectories in Figure 1C, the improvement in trajectory consistency is not clear (day 2 seems to be more consistent than day 8). If the improvement in trajectory consistency is less pronounced than the increase in fast speed consistency, the authors should make significant modifications in the way they present their behavioral data and acknowledge that their physiological changes could explain either trajectory consistency, increased (fast) speed consistency, increase speed or a mixture of the three.

– Increase in speed consistency does not necessarily imply an increase in speed. The striking increase in reaching speed with learning (Figure 1D) is just shown for one animal. The authors should show it for their 6 animals. If the 6 animals show an increase in movement speed this is a point that should be discussed throughout the manuscript, especially in light of the many works linking dorsal striatum and movement speed.

– In figure 1e, the authors showed in grey the individual speed profile correlation of 6 rats and the mean+sem. Something is quite wrong with the mean trace. In the first days, the mean should be much higher, close to 0.8. Currently, its first value is between the 5th and 6ht values. Second, on a statistical point of view, using mean and SEM is meaningless when n=6. The median would make more sense and there is no need for error bars as the entire dataset is shown. Once the group representation is corrected, taking into account that the y-axis in e) is cut at 0.6, it will be clear that the increase in speed profile consistency is far from impressive.

– To demonstrate that the behavior is automatic, in the new pellet location task, the authors used trajectory correlation to quantify behavior, that is, a different metric than in figure 1. This inconsistency in behavioral metrics across two related figures (Figure 1 and Figure S1) raises the question of whether the lack of behavioral change shown in Figure S1 (which the authors use to claim automaticity) is robust when looking at speed (either absolute speed or consistency). Added to the fact that this experiment was only performed on two animals this part was really not convincing. Anyway, the authors should also examine whether movement speed is affected by the relocation of the pellet.

2. Validate LFP recordings

To study M1 -DLS functional coupling, the authors, in some of their analyses, used striatal LFP to compute M1 and DLS coherency. When introducing their result section, the authors stated that "within the corticostriatal network, theta coherence (4-8Hz) has been previously shown to reflect coordinated population spiking activity8,9,34 " (Line 128). The authors should mention recent works showing major volume conduction in the striatum in this frequency range (Lalla et al., 2017) and in the γ range (Carmichael et al., 2017), as the authors seem to be aware of this potential confound (in Lemke et al. NN, 2019, relevant works are cited). Sleep rhythms are well known to be controlled by the thalamocortical systems and the LFP' sources to be in the cortex (Kandel and Buzsaki 1997). The lack of organization of the input on striatal neurons along with their radial somatodendritic shape makes the striatum a poor candidate to generate fields that can summate and be recorded extracellularly. Because the striatal recording sites in the present study are located just below the cortex it is very likely that most, if not all the striatal LFPs is volume-conducted from neighboring cortical sources. The authors said they used common-average referencing to limit volume conduction but this will not fix the problem. Indeed, subtracting two oscillatory signals with the same phase and frequency, but different amplitude (as it can happen in the striatum due to the passive attenuating effects of the brain tissue on LFPs) will result in an oscillatory signal with a preserved rhythmicity. The authors should look carefully at the work of Carmichael and collaborator (2017). In this study, the authors showed that striatal LFPs amplitude were slowly decreasing as the striatal electrodes were further away from the cortex. Carmichael also showed that striatal spiking modulation by striatal LFPs is not a criterion for local generation of the field (all it shows is that striatal units are modulated by cortical rhythms which is expected as the cortex provides strong excitation to the striatal neurons). If the authors want to make a claim about striatal LFPs beeing local then they need to show that their striatal LFPs (referenced against their cereball screw) are not progressively decreasing away from the cortex. It is an important issue that can not be ignored by the authors.

– Monosynaptically connected pairs and cross-correlograms (CCGs). The authors claim to identify monosynaptically connected neuron pairs from CCGs. Previous works (see for instance Bartho et al. 2004) have shown extremely sharp peaks in CCG of putatively connected neurons in the cortex and hippocampus but something is clearly different in the CCGs shown here. Indeed, it is striking that the CCG peaks shown are very smoothed (it is more a wave than a peak). In fact, this wave crosses the center of CCG and seems significant in the positive time bins, suggesting that striatal neurons fire before cortical neurons, which makes little sense. It is surprising that the authors did not mention in their result section the potential alternative mechanisms explaining such "peaks" that spread until positive CCG values. Indeed, it is well known that there are alternative explanations for these observations, such as indirect polysynaptic partners as well as common ("third party") input or slower co-modulation (see Brody CD (1999) Correlations without synchrony, and several papers by Asohan Amarasingham and colleagues). Amarasingham et al. (2012), J. Neurophysiology, talks about interval jitter, why it's relevant to separating fast from slow comodulation and explains the history of these problems. In this regard, the shuffling method briefly mentioned by the authors in the method section is unclear and does not seem to address properly the issue of separating fast and slow comodulation (see Amarasingham 2012).

– While M1 RM significance passed the arbitrary α of 0.05, it is a dramatically weaker effect than that seen in DLS. I would ask the authors please comment on this difference. Additionally, both Δ and SO strongly modulate M1, and **to a greater extent than any of the other effects mentioned in the text!** Please point this out to your readers and offer an interpretation. Do these results argue that in M1 – unlike DLS- the spindles not the main contributor? Or if nothing else, that the modulation is non-specific? (I think the former. this may actually help the authors in their attempts to dissociate m1 from dls sleep processes). The spindle story is great, but these results change the interpretation should not be buried.

3. Neural Trajectories

There were 3 major concerns raised 1)how the trajectories are composed, 2) the validity of the predicted trajectory and 3) the comparison of the trajectories:

1) The pc plots represent trial-averaged activity. Although no reach dynamics are given in this paper, a very similar study by the same authors shows that early on, the variability in duration of the outward, and outward+grasping, are very high compared to late in learning. Whatever 'reach signal' is present, this variability will cause the trial-averaged values to be greatly reduced. It is recommended that the authors address this by: (a) time-warping individual trials to overcome this and (b) doing this for only the outward trajectory, as their previous paper shows that so much of the change in duration due to learning is the result of improved grasping ( e.g the difference between PT and RO in Lemke 2019).

2) It is also unclear if the authors are trying to say in Figure 2 "Offline increases in functional connectivity predict the emergence of low-dimensional cross area neural dynamics during behavior" The authors should disambiguate cross area dynamics from within area dynamics. Previous work has shown that there is little to no goal-related movement activity early on in training. More importantly, the authors themselves show this in their 2019 paper that although there is some Day 1 m1 spiking modulation related to the reach, the psth for striatum is flat psth (previous comments on variability of timing still apply). They also show some reach-aligned power increase in 4-8 Hz in M1, but nothing in striatum. Given the small dynamics(?) that might be in the striatum, it is actually impressive that there is as much similarity between DLS and predict as there is. Can the authors show that this is a lack of a cross-area relationship, not just a lack of signal – which would preclude even the possibility of that relationship. Potentially they could scale with the power of the signal – amplitude or % explained variability of the first PC's, or just plot avg dms modulation in b. Otherwise, the cross area dynamics argument is not compelling. This same problematic facet comes into play later when determining reach modulation (which was done across all days).

3) Finally, how do you quantify quality of prediction? "The predictive ability of these models was assessed by calculating the correlation between the actual neural trajectories and the predicted trajectories." This explanation lacks vital details about the comparison and the model itself. Moreover, these are time-varying 2-D vectors, where one offset or delay early on can propagate throughout the trajectory, even though the trajectories would otherwise be identical. Much work has been done on this in recent years, as this is a difficult question to tackle. Euclidean distances of a time-warped distribution or point distribution models might help. Regardless, more detail is needed.

4. Clarify data presentation:

– Figure 2c – The authors show an interesting temporal dynamic here, and there are hints that online DEcorrelation may contribute to learning (for which there is some evidence) as well as offline increases in coherence. This may be just a happenstance of this specific example, but 2c makes it clear that showing the trends across time, rather than in sum as in Figure 2d, would be helpful. Could the authors please add a figure like 2c, but across all 6 animals? It would be nice to see this possibility of online changes at least briefly discussed as well.

– Fig4 sup2, etc – The cited conduction delay between m1 and DS is very specific: 5-7ms (and with a decrease beyond baseline in the 1-3ms range). The authors should redo their analysis using 1-10ms to 5-10ms, or provide a clearly compelling rationale for including 1ms latencies.

Moreover, it is unclear why the x scale is +/- 75ms? this seems quite large and belies the bigger question of why the distributions are so large, especially for data that should be normalized (subtracting out baseline correlations, and thus should be sharper). The largely symetic and broad increase, extending before 0 lag, is worrisome. If it is a function sleep vs wake, can connectivity be determined only during wake?

AP5 effects:

– The presentation and description of the results are confusing and incomplete. The experimental design could not be interpreted by reading the result section or looking at the figure 1 (panel f and g). In the method section, the authors wrote that " In the first five days of training, we infused three rats with AP5 and three rats with saline, for the second five days, we switched the infusion, i.e., animals that received AP5 in the first five days, received saline for the second five days, and vice-versa". First, this sentence should be included in the main text. Second, why did the authors show only one animal in figure 1f ? they should show the six rats.

– I could not make sense of the analytical logic in panel g. The authors correlated the trial by trial speed profiles during training with the average speed profile at the end of training, saline, or AP5 sessions. During "control" training (saline and learning cohorts), as the behavior changes, there should be lower correlation at the beginning of training and higher correlation at the end. In addition when the speed profile does not change (early AP5 or late learning cohort) there could be higher correlations. It is not clear to me how this measurement can be useful as it may vary a lot during learning and could also be different depending on the type of behavioral plateau.

– Also it is unclear whether the effect of AP5 on day-to-day correlation should be the same when injected since day1 or after 5 days of practice under saline condition. The data shown for one animal seems to show a clear decrease in speed profile correlation during AP5. What do the authors make of such effect ? It seems to indicate that AP5 not only block automatization but also impair performance. Is a similar effect observed when AP5 given after 5 days of saline ? The effect of AP5 on movement speed should be shown.

– Histology is absent. At a minimum, ascertaining how well layer 5 was targeted across animals should be included.

– Reviewers were confused by sleep nomenclature:

Figure 2a – it was not obvious that sleep 3 = next day's sleep 1. can you include something to the effect of "sleep3 = day n+1's sleep 1" just to help make it perfectly clear to the readers?

Another commented that "pre-sleep" and "post-sleep" nomenclature very confusing. These aren't used to define time periods before and after a sleeping episode, but instead are periods of "sleep" (although there can be wake states mixed in) before and after training. "Pre-training sleep" and "post-training sleep" would be much more clear.

5. Reach modulation across days

– Reach Modulation appears to be determined across all days? This assumes that each unit is maintained across 8 days- evidence supporting this needs to be provided. More importantly though, the potential for this to change the DLS results is remarkable. As previously stated, the authors showed very little reach related activity on day 1. Why not determine these values for day 1-2 vs day 7-8? or better yet, as somewhat suggested by the authors, can they predict what neurons will become RM over time?

– It seems that the day-to-day correlation measurement allows the authors to have more data points per rat. However, these measurements for a given animal are not independent. Unfortunately, pseudoreplication is a major issue in neuroscience and behavioral studies more broadly (S.E Lazic's "The problem of pseudoreplication in neuroscientific studies: is it affecting your analysis?" BMC Neuroscience, 2010, highlights some things to worry about in this regard). The authors should avoid this in this figure and other neurophysiological analyses.

6. NREM sleep

– It would be interesting to know how many times the animals entered NREM sleep between the post-training sleep day N and pre-sleep of subsequent day N+1. Are these between day changes depended upon the total amount of sleep that the animals got?

– pg6 line 192. In order to ascertain that NREM > others, it would be useful to see this relationship in unconnected pairs. e.g. control for the possibility that NREM is always highest.

– Since all of the data is from a specific subsection of these offline states (i.e. – NREM sleep) it would be good to discuss their findings within the context of what is known/unknown about plasticity processes during offline wake states or other sleep periods.

7. Topics to clarify/discuss

– Page 7 Line 234 – "In contrast, DLS units did not increase in modulation during either δ waves (Supplemental Figure 5) or slow oscillations (Supplemental Figure 6) after training." This statement is only partially correct. I applaud the authors giving, and showing, the same treatment for all 3 types of data. Many would gloss over this. DLS nonRM weak input demonstrated a stronger effect than M1 RM, and SO demonstrated a trend. Some will find this juxtaposition – task related-spindle, unrelated-other to be informative, particularly as information itself can be more strongly represented through increasing the signal or increasing the discernability from not-signal. There is considerable work, including from the authors, showing the importance of other elements of sleep. Throughout the paper, SO and δ aren't negligible contributors. Rather than glaze over this for the purpose of a throughline, please expand upon this for a more complete view.

– The discussion is somewhat lacking, being just a summary of the findings. In particular, I would like to see the authors put these results into context with the broader models of Tononi (SHY) and their own work, e.g. Kim 2019 wherein they focus on SO and δ in a very similar task – but which are largely thrown under the bus here.

– It is unclear why "automaticity" is a beneficial outcome here for this task. Does it result in greater pellet retrieval success (no data presented on this point) or does it free up or allow reorienting of cognitive resources (this is indirectly alluded to in the Discussion)?

– The argument structure in the Intro has a few holes in it.

a) Specifically the paragraph covering lines 42-53. "Currently, our understanding of how sleep impacts distributed brain networks is largely derived from the systems consolidation theory, where it has been shown that coordinated activity patterns across hippocampus and cortex lead to the formation of stable long-term memories in cortex that do not require the hippocampus 23-25. Notably, whether sleep impacts the connectivity across hippocampus and cortex has not been established. Therefore, one possibility is that, in the network, we similarly observe coordinated cross-area activity patterns during sleep but do not find evidence for the modification of corticostriatal connectivity during offline periods. Alternatively, it is possible that we find evidence that cross-area activity patterns during sleep modify the connectivity between cortex and striatum and impact network activity during subsequent behavior."

b) The first two sentences describe the role of sleep in systems consolidation theory and "coordinated activity patterns" and "connectivity" across hippocampus and cortex. The third and fourth sentences then jump to two alternative speculations about "the network" (which is undefined), and the role of sleep in modifying corticostriatal connectivity patterns. It is not clear how these two things are linked. This link should be explicitly stated.

– Offline is used in two different ways here. Offline behavioral states include any sleeping or waking state outside of training. Then there are online (between pre- and post-sleep on the same day) and offline (between post-sleep of day N and pre-sleep of subsequent day N+1) changes in neural activity between these offline behavioral states.

– The authors need to better define what they mean by slow oscillations (Figure 7, Figure 4e). Cortical slow oscillations are classically described as regular up and down states transition, with sleep spindles occurring during the up states (Buzsaki, rhythms of the brain p 197). When the authors look at the spiking modulation by slow oscillations and spindles (Figure 4e) or the temporal proximity between slow oscillations and spindles (Figure 7), I doubt the authors are referring to this classical up/down slow oscillations. Are the slow oscillations the authors refer to equivalent to K complexes? If yes, this could be mentioned.

*Reviewer #1:*

Lemke and co-authors present a thorough interrogation of the role of sleep in motor memory consolidation and performance, a clear continuation of the lab's focus. Multiple techniques are used to arrive at the conclusions, which are largely robust and arrived at through sound scientific methods. It is a rather beefy and ambitious paper, and for that and the polish, the authors should certainly be commended. Based on the findings and the quality of the work, I am eager to see the work brought to publication but some critical points must be addressed before continuing the discussion.

Figure 2c – The authors show an interesting temporal dynamic here, and there are hints that online DEcorrelation may contribute to learning (for which there is some evidence) as well as offline increases in coherence. This may be just a happenstance of this specific example, but 2c makes it clear that showing the trends across time, rather than in sum as in Figure 2d, would be helpful. Could the authors please add a figure like 2c, but across all 6 animals? It would be nice to see this possibility of online changes at least briefly discussed as well.

Figure 3 – I appreciate the authors' attempts here. Previously they showed an increase in within-area spiking and 4-8hz dynamics as learning progressed, so this work is a natural progression. However, I have three major reservations concerning (1)how the trajectories are composed, (2) the validity of the predicted trajectory and (3) the comparison of the trajectories:

1) The pc plots represent trial-averaged activity. Although no reach dynamics are given in this paper, a very similar study by the same authors shows that early on, the variability in duration of the outward, and outward+grasping, are very high compared to late in learning. Whatever 'reach signal' is present, this variability will cause the trial-averaged values to be greatly reduced. I would recommend a) time-warping individual trials to overcome this and b) doing this for only the outward trajectory, as their previous paper shows that so much of the change in duration due to learning is the result of improved grasping ( e.g the difference between PT and RO in Lemke 2019). If the authors choose to include full trajectories, or better yet, outward and inward, this would be fine as well, but as the limb trajectories are only outward, I would compare apples to apples.

2) It is also unclear if the authors are trying to say in Figure 2 / "Offline increases in functional connectivity predict the emergence of low-dimensional cross area neural dynamics during behavior" In particular, I would disambiguate cross area dynamcs from within area dynamics. David Robbe and others have shown data suggesting that there is little to no goal-related movement activity early on in training. More importantly, the authors themselves show this in their 2019 paper that although there is some Day 1 m1 spiking modulation related to the reach, the psth for striatum is flat psth (previous comments on variability of timing still apply). They also show some reach-aligned power increase in 4-8 Hz in M1, but nothing in striatum. Given the small dynamics(?) that might be in the striatum, I'm actually impressed that there is as much similarity between DLS and predict as there is. Thus, I ask the authors to show that this is a lack of a cross-area relationship, not just a lack of signal – which would preclude even the possibility of that relationship. Potentially they could scale with the power of the signal – amplitude or % explained variability of the first PC's, or just plot avg dms modulation in b. Otherwise, I don't quite buy the cross area dynamics argument. This same problematic facet comes into play later when determining reach modulation (which was done across all days).

3) Finally, how do you quantify quality of prediction? "The predictive ability of these models was assessed by calculating the correlation between the actual neural trajectories and the predicted trajectories." This explanation lacks vital details about the comparison and the model itself. Moreover, these are time-varying 2-D vectors, where one offset or delay early on can propagate throughout the trajectory, even though the trajectories would otherwise be identical. Much work has been done on this in recent years, as this is a difficult question to tackle. Euclidean distances of a time-warped distribution or point distribution models might help. Regardless, more detail is needed.

Figure 4 sup2, etc – The cited conduction delay between m1 and DS is very specific: 5-7ms (and with a decrease beyond baseline in the 1-3ms range). I would ask the authors to redo their analysis using 1-10ms to 5-10ms, or provide a clearly compelling rationale for including 1ms latencies.

Moreover, it is unclear to me why is the x scale +/- 75ms? this seems quite large and belies the bigger question of why the distributions are so large, especially for data that should be normalized (subtracting out baseline correlations, and thus should be sharper). The largely symmetric and broad increase, extending before 0 lag, is worrisome. I would expect even the average data to look more like 6c right, or highlighted in the differences between sup3c top/bottom.

If it is a function sleep vs wake, can connectivity be determined only during wake?

Figure 4 – I would appreciate a differentiation of spn's / fsi's. even just baseline firing rate. although I assume most of the striatal neurons are spn's, it would be good to know a) if the ratio was above chance and b) if the findings generalized to both populations.

pg6 line 192. In order to ascertain that NREM > others, it would be useful to see this relationship in unconnected pairs. e.g. control for the possibility that NREM is always highest.

Reach Modulation appears to be determined across all days? This assumes that each unit is maintained across 8 days (I won't make a stink about this, but if this assertion is to be made, I should be substantiated). More importantly though, the potential for this to change the DLS results is remarkable. As previously stated, the authors showed very little reach related activity on day 1. Why not determine these values for day 1-2 vs day 7-8? or better yet, as somewhat suggested by the authors, can they predict what neurons will become RM over time?

While M1 RM significance passed the arbitrary α of 0.05, it is a dramatically weaker effect than that seen in DLS. I would ask the authors please comment on this difference. Additionally, both Δ and SO strongly modulate M1, and to a greater extent than any of the other effects mentioned in the text! Please point this out to your readers and offer an interpretation. Do these results argue that in M1 – unlike DLS- the spindles not the main contributor? Or if nothing else, that the modulation is non-specific? (I think the former. this may actually help the authors in their attempts to dissociate m1 from dls sleep processes). The spindle story is great, but these results change the interpretation should not be buried.

Page 7 Line 234 – "In contrast, DLS units did not increase in modulation during either δ waves (Supplemental Figure 5) or slow oscillations (Supplemental Figure 6) after training." This statement is only partially correct. I applaud the authors giving, and showing, the same treatment for all 3 types of data. Many would gloss over this. DLS nonRM weak input demonstrated a stronger effect than M1 RM, and SO demonstrated a trend. Some will find this juxtaposition – task related-spindle, unrelated-other to be informative, particularly as information itself can be more strongly represented through increasing the signal or increasing the discernability from not-signal. There is considerable work, including from the authors, showing the importance of other elements of sleep. Throughout the paper, SO and δ aren't negligible contributors. Rather than glaze over this for the purpose of a throughline, please expand upon this for a more complete view.

The discussion is somewhat lacking, being just a summary of the findings. In particular, I would like to see the authors put these results into context with the broader models of Tononi (SHY) and their own work, e.g. Kim 2019 wherein they focus on SO and δ in a very similar task – but which are largely thrown under the bus here.

Methods – thank you for indicating the data for which each method applies. It really helps the reader.

histology is absent but I do not see this as an absolutely critical element. I will say that ascertaining how well layer 5 was targeted across animals would be very helpful

*Reviewer #2:*

This is a well written, novel, technically sound and timely paper. Addressing the following issues could strengthen it further.

It is unclear why "automaticity" is a beneficial outcome here for this task. Does it result in greater pellet retrieval success (no data presented on this point) or does it free up or allow reorienting of cognitive resources (this is indirectly alluded to in the Discussion)?

The argument structure in the Intro has a few holes in it. Specifically the paragraph covering lines 42-53.

"Currently, our understanding of how sleep impacts distributed brain networks is largely derived from the systems consolidation theory, where it has been shown that coordinated activity patterns across hippocampus and cortex lead to the formation of stable long-term memories in cortex that do not require the hippocampus 23-25. Notably, whether sleep impacts the connectivity across hippocampus and cortex has not been established. Therefore, one possibility is that, in the network, we similarly observe coordinated cross-area activity patterns during sleep but do not find evidence for the modification of corticostriatal connectivity during offline periods. Alternatively, it is possible that we find evidence that cross-area activity patterns during sleep modify the connectivity between cortex and striatum and impact network activity during subsequent behavior."

The first two sentences describe the role of sleep in systems consolidation theory and "coordinated activity patterns" and "connectivity" across hippocampus and cortex. The third and fourth sentences then jump to two alternative speculations about "the network" (which is undefined), and the role of sleep in modifying corticostriatal connectivity patterns. It is not clear how these two things are linked. This link should be explicitly stated.

I found the "pre-sleep" and "post-sleep" nomenclature very confusing. These aren't used to define time periods before and after a sleeping episode, but instead are periods of "sleep" (although there can be wake states mixed in) before and after training. "Pre-training sleep" and "post-training sleep" would be much more clear.

Offline is used in two different ways here. Offline behavioral states include any sleeping or waking state outside of training. Then there are online (between pre- and post-sleep on the same day) and offline (between post-sleep of day N and pre-sleep of subsequent day N+1) changes in neural activity between these offline behavioral states.

It would be interesting to know how many times the animals entered NREM sleep between the post-training sleep day N and pre-sleep of subsequent day N+1. Are these between day changes depended upon the total amount of sleep that the animals got?

Also, since all of their data is from a specific subsection of these offline states (i.e. – NREM sleep) it would be good to discuss their findings within the context of what is known/unknown about plasticity processes during offline wake states or other sleep periods.

*Reviewer #3:*

In this manuscript, Lemke and collaborators examined whether offline changes in the functional coupling between the primary motor cortex (M1) and the dorsolateral striatum (DLS) contribute to the automatization of reaching movements in rats. The authors performed perturbation of striatal activity and simultaneous multi-unit/LFP recordings in M1 and DLS during sleep/rest recording sessions before and after reaching sessions. The authors report behavioral impairment following blocking of striatal NMDA receptors during offline periods. They also show the results of analyses congruent with the idea that there is an increased functional coupling between M1 and DLS observed during sleep, which parallels the increase in automaticity. They also point at sleep spindles as a critical period of corticostriatal plasticity.

A majority of studies have examined corticostriatal activity during behavior and the focus of this study on what's going during sleep/rest periods is very interesting, especially taking into account the effect of sleep on consolidation of motor skills. The electrophysiological recordings performed are extremely challenging experiments and consequently, the data generated are extremely rich. Still, large-scale unit activity and sleep-related LFPs rhythms are notoriously tricky to analyze, especially in such different structures as M1 and DLS. In this context, I found that some of the main claims were a bit hasty considering the inherent limitations of the analyses performed (extracellular recordings are blind to many of the underlying mechanisms and multiple confounds that could account for increased coordination between LFPs and spiking patterns across brain regions). In addition, the behavioral analyses performed were statistically problematic and did not address the potential interaction/confound between automaticity and speed/vigor, which is relevant to the corticostriatal function. In conclusion, IMO, this is an impressive experimental work with potentially interesting results on a topic that has not been very well studied, but the manuscript should be significantly improved in several key points.

1. Trajectory consistency and movement speed

Throughout the manuscript, the authors define automaticity has an increased consistency in reaching trajectory (abstract, line 23-26, line 86) but the main measurement used in figure 1 is the correlation of the velocity profile (which actually shows a strong increase in speed during learning). This is problematic for several reasons:

-An inattentive reader may think that such that the y-axis label "reach correlation" in panels e, f, and g of figure 1 is referring to the correlations of the trajectories shown in c.

-This choice raises the question of why the authors quantified behavior by looking at the consistency of the speed profile rather than directly through trajectories correlation. Looking at the traces of the trajectories in Figure 1C, the improvement in trajectory consistency is not clear (day 2 seems to be more consistent than day 8). If the improvement in trajectory consistency is less pronounced than the increase in fast speed consistency, the authors should make significant modifications in the way they present their behavioral data and acknowledge that their physiological changes could explain either trajectory consistency, increased (fast) speed consistency, increase speed or a mixture of the three.

– Increase in speed consistency does not necessarily imply an increase in speed. The striking increase in reaching speed with learning (Figure 1D) is just shown for one animal. The authors should show it for their 6 animals. If the 6 animals show an increase in movement speed this is a point that should be discussed throughout the manuscript, especially in light of the many works linking dorsal striatum and movement speed.

– In figure 1e, the authors showed in grey the individual speed profile correlation of 6 rats and the mean+sem. Something is quite wrong with the mean trace. In the first days, the mean should be much higher, close to 0.8. Currently, its first value is between the 5th and 6ht values. Second, on a statistical point of view, using mean and SEM is meaningless when n=6. The median would make more sense and there is no need for error bars as the entire dataset is shown. Once the group representation is corrected, taking into account that the y-axis in e) is cut at 0.6, it will be clear that the increase in speed profile consistency is far from impressive.

– To demonstrate that the behavior is automatic, in the new pellet location task, the authors used trajectory correlation to quantify behavior, that is, a different metric than in figure 1. This inconsistency in behavioral metrics across two related figures (Figure 1 and Figure S1) raises the question of whether the lack of behavioral change shown in Figure S1 (which the authors use to claim automaticity) is robust when looking at speed (either absolute speed or consistency). Added to the fact that this experiment was only performed on two animals this part was really not convincing. Anyway, the authors should also examine whether movement speed is affected by the relocation of the pellet.

2. AP5 effects.

– The presentation and description of the results are confusing and incomplete. I could not understand the experimental design by reading the result section or looking at the figure 1 (panel f and g). In the method section, the authors wrote that " In the first five days of training, we infused three rats with AP5 and three rats with saline, for the second five days, we switched the infusion, i.e., animals that received AP5 in the first five days, received saline for the second five days, and vice-versa". First, this sentence should be included in the main text. Second, why did the authors show only one animal in figure 1f ? they should show the six rats.

– I could not make sense of the analytical logic in panel g. The authors correlated the trial by trial speed profiles during training with the average speed profile at the end of training, saline, or AP5 sessions. During "control" training (saline and learning cohorts), as the behavior changes, there should be lower correlation at the beginning of training and higher correlation at the end. In addition when the speed profile does not change (early AP5 or late learning cohort) there could be higher correlations. It is not clear to me how this measurement can be useful as it may vary a lot during learning and could also be different depending on the type of behavioral plateau.

– Also it is unclear whether the effect of AP5 on day-to-day correlation should be the same when injected since day1 or after 5 days of practice under saline condition. The data shown for one animal seems to show a clear decrease in speed profile correlation during AP5. What do the authors make of such effect ? It seems to indicate that AP5 not only block automatization but also impair performance. Is a similar effect observed when AP5 given after 5 days of saline ? The effect of AP5 on movement speed should be shown.

– It seems that the day-to-day correlation measurement allows the authors to have more data points per rat. However, these measurements for a given animal are not independent. Unfortunately, pseudoreplication is a major issue in neuroscience and behavioral studies more broadly (S.E Lazic's "The problem of pseudoreplication in neuroscientific studies: is it affecting your analysis?" BMC Neuroscience, 2010, highlights some things to worry about in this regard). The authors should avoid this in this figure and other neurophysiological analyses.

3. Striatal LFPs. To study M1 -DLS functional coupling, the authors, in some of their analyses, used striatal LFP to compute M1 and DLS coherency. When introducing their result section, the authors stated that "within the corticostriatal network, theta coherence (4-8Hz) has been previously shown to reflect coordinated population spiking activity8,9,34 " (Line 128). I find it a bit unfair from the authors not to mention recent works showing major volume conduction in the striatum in this frequency range (Lalla et al., 2017) and in the γ range (Carmichael et al., 2017), as the authors seem to be aware of this potential confound (in Lemke et al. NN, 2019, relevant works are cited). Sleep rhythms are well known to be controlled by the thalamocortical systems and the LFP' sources to be in the cortex (Kandel and Buzsaki 1997). The lack of organization of the input on striatal neurons along with their radial somatodendritic shape makes the striatum a poor candidate to generate fields that can summate and be recorded extracellularly. Because the striatal recording sites in the present study are located just below the cortex it is very likely that most, if not all the striatal LFPs is volume-conducted from neighboring cortical sources. The authors said they used common-average referencing to limit volume conduction but this will not fix the problem. Indeed, subtracting two oscillatory signals with the same phase and frequency, but different amplitude (as it can happen in the striatum due to the passive attenuating effects of the brain tissue on LFPs) will result in an oscillatory signal with a preserved rhythmicity.

The authors should really look carefully at the work of Carmichael and collaborator (2017). In this study, the authors showed that striatal LFPs amplitude were slowly decreasing as the striatal electrodes were further away from the cortex. Carmichael also showed that striatal spiking modulation by striatal LFPs is not a criterion for local generation of the field (all it shows is that striatal units are modulated by cortical rhythms which is expected as the cortex provides strong excitation to the striatal neurons). If the authors want to make a claim about striatal LFPs being local then they need to show that their striatal LFPs (referenced against their cerebral screw) are not progressively decreasing away from the cortex. It is an important issue that can not be ignored by the authors.

4. Monosynaptically connected pairs and cross-correlograms (CCGs). The authors claim to identify monosynaptically connected neuron pairs from CCGs. Previous works (see for instance Bartho et al. 2004) have shown extremely sharp peaks in CCG of putatively connected neurons in the cortex and hippocampus but something is clearly different in the CCGs shown here. Indeed, it is striking that the CCG peaks shown are very smoothed (it is more a wave than a peak). In fact, this wave crosses the center of CCG and seems significant in the positive time bins, suggesting that striatal neurons fire before cortical neurons, which makes little sense. I am surprised that the authors did not mention in their result section the potential alternative mechanisms explaining such "peaks" that spread until positive CCG values. Indeed, it is well known that there are alternative explanations for these observations, such as indirect polysynaptic partners as well as common ("third party") input or slower co-modulation (see Brody CD (1999) Correlations without synchrony, and several papers by Asohan Amarasingham and colleagues). Amarasingham et al. (2012), J. Neurophysiology, talks about interval jitter, why it's relevant to separating fast from slow comodulation and explains the history of these problems. In this regard, the shuffling method briefly mentioned by the authors in the method section is unclear and does not seem to address properly the issue of separating fast and slow comodulation (see Amarasingham 2012).

[Editors' note: further revisions were suggested prior to acceptance, as described below.]

Thank you for resubmitting your work entitled "Coupling between motor cortex and striatum increases during sleep over long-term skill learning" for further consideration by *eLife*. Your revised article has been evaluated by Michael Frank (Senior Editor) and a Reviewing Editor.

The manuscript has been improved but there are some remaining issues that need to be addressed, as outlined below:

1. A graph showing the effects of AP5 on speed, for all 6 mice should be shown in the main figure in a single plot with a fixed scale. All of Reviewer #3's requests for this figure should be included, including statistics.

2. This figure is requested to address potential confounds of effects on vigor that are currently interpreted as effects on motor learning. Depending on how conclusive the effects of AP5 on speed are, the authors might need to revise their language throughout the paper, if an effect of movement vigor cannot be rules out.

3. A paragraph should be added to the Discussion explaining the potential confounds of effect on movement vigor on the study's results.

Although these points came from Reviewer 3, in consultation the other Reviewers agreed these points above are important to address.

*Reviewer #1:*

The reviewed document is much more coherent, with very clear and intuitive figures in addition to a high degree of rigor. I have nothing to add at this time.

*Reviewer #2:*

Comments have been satisfactorily addressed.

*Reviewer #3:*

The authors have made a significant effort to reply to most of my comments and those of the other reviewers. However one of the major points that I had asked has not been addressed and another related one is not addressed properly. Those points were critical for the authors claim that "our results provide evidence that sleep shapes cross-area coupling required for skill learning" (last sentence abstract, maybe a "is" is missing?).

Specifically I asked the authors to report the effect of AP5 injection on movement speed. They responded that they provide the effect of AP5 on the speed profile correlation but clearly this is not what I asked. I don't think the authors misunderstood me because in reply to another comment about movement speed (my original comment :"-Increase in speed consistency does not necessarily imply an increase in speed. The striking increase in reaching speed with learning (Figure 1D) is just shown for one animal. The authors should show it for their 6 animals. If the 6 animals show an increase in movement speed this is a point that should be discussed throughout the manuscript, especially in light of the many works linking dorsal striatum and movement speed"), the authors did provide in their supplementary figure 3, plots showing that movement speed increases in the 6 animals during training. Thus it is unclear why the authors did not show the effect of AP5 on movement speed.

My request is really fundamental in regard of the main claim of the author (requirement for skill learning) because it is possible that AP5 decreases not just the trial-by-trial speed profile correlation (what the authors use as a definition for skill) but also the general speed of movements which is not necessarily related to skill learning but could have a motivational origin (see work of the Galea lab). If this was the case, the authors would need to seriously consider that the changes in corticostriatal connectivity could primarily reflect altered vigor or motor motivation which would be in agreement with several works in the field (Rob Turner lab, Dudman lab, Robbe lab, or even ideas on motor motivation by Josh Berke, one of the authors of this study).

This point is even more important as the authors, in the introduction or discussion, tend to write definitive sentences on a well-demonstrated role of cortico-striatal connection in motor skills while most of the references cited do not disambiguate the vigor or motivation confound (e.g., Kupferschmidt et al., Dang et al., Costa et al., Yin et al. …). This is quite misleading.

Moreover, when looking at the result of the AP5 experiment on the speed profile correlation across the 6 animals (supp Figure 4), the results are far from convincing. There are only 3 animals in each condition (3 in AP5-Saline and 3 in Saline AP5) and in each of the conditions, the result is not clear in at least one animal. There are also weird day by day drastic changes that make these results not so reliable. Moreover, the authors keep changing the y axis scale in a way that makes small improvements look big. Ethically speaking this is not appropriate, especially in such a journal as *eLife*. Neither is appropriate to put in the main figures 1d and 1e the best example out of 6 animals. The authors should integrate into these panels the results of the other animals. In addition to being fairer to the data, this would remove two supplementary figures.

Thus, the main figure 1 should show 1) the effect of AP5 on speed profile correlation on the 6 animals in the same graph with the same scale (Figure 1e); 2) the effect of AP5 on max speed on the 6 animals (new small panel) and the fig1d for all the animals. In addition, the statistics to examine the effect of AP5 should be done using paired comparison of the correlation and speed values not on their change. The usage of the changes in panel 1f is unfair because it is not clear what should be the effect of AP5 after saline (decrease or plateau?) I am afraid that the authors designed their statistical test in a way that favored their hypothesis.

In conclusion, while I do think this paper should be published in *eLife*, I am convinced that a fairer presentation and analyses of key experiments in figure 1 are required. Clearly, the effect of AP5 on skill learning is not as clear-cut as stated by the authors (and choosing to only show in the main figure the single best animal is not appropriate). In addition, an effect on movement speed is still highly likely (based on one animal) and should be reported for the 6 animals. Thus the main behavioral function of the interesting neurophysiological changes could be related to vigor/motor motivation, not skill learning. All this critical information should not be hidden but rather openly disclosed and discussed such as the reader can make up its mind.

[Editors' note: further revisions were suggested prior to acceptance, as described below.]

Thank you for resubmitting your work entitled "Coupling between motor cortex and striatum increases during sleep over long-term skill learning" for further consideration by *eLife*. Your revised article has been evaluated by Michael Frank (Senior Editor) and a Reviewing Editor.

*Reviewer #3:*

I asked the authors to show the data for the 6 animals in which they compared AP5 and Saline striatal injection on the speed correlation profile. The authors have managed to plot the data in a way in which it is impossible to know which rat is which (all the points are gray, no running lines between the points). They plotted the mean of these data which is meaningless from a statistical viewpoint (n=6).

I also asked the authors to show on the same main figure the effect of AP5 on the max speed (to test the effect on vigor). The authors have put those results in a supplementary figure. The authors claim that AP5 has only an effect on the speed correlation profile not on max speed. But this is clearly due to their biased statistic in which they only look at the difference between the first and last sessions. Indeed in figure 1e AP5 only reduces the speed correlation on day 5 but there is no effect from day 1 to day4. Juxtaposing the effect of AP5/Saline on max speed (now Figure 1 S5) and speed correlation ( figure 1e bottom) clearly shows that the effects are strikingly similar. In fact, an unbiased treatment of the data (using the entire profile and permutation/bootstrapping) would probably show that the effect of AP5 on max speed is more pronounced than on speed correlation.

The authors seem to conclude that the AP5 effect is different on speed profile correlation and max speed because their statistical comparison with saline is significant in one case and insignificant in the other case.

However, comparing p values is meaningless statistically. This is an important issue that has been subject to publication in highly visible journals (https://www.nature.com/articles/nn.2886).

The authors should have compared the effect themselves which again should be done using permutation/bootstrapping on the entire profile (not last/first).

Thus I maintain that impartial analyses of the data cannot disentangle whether the DLS is primarily contributing to learning the accurate movement or contribute to the increased vigor which is driven by a motivational aspect. I am glad the authors acknowledge it in their discussion. However, most readers will go quickly through the title and abstract (and maybe introduction) and will probably cite this paper as additional evidence for a critical role of the striatum in motor skill learning which in my opinion is misleading.

---

## [Author Response]

Essential revisions:The reviewers felt that the topic of the paper is very interesting and the study is original and creative. However, a number of issues were raised about clarity of the presentation, data analysis, and interpretation of the results. The concerns fall under the following categories:1. Trajectory consistency and movement speedThroughout the manuscript, the authors define automaticity has an increased consistency in reaching trajectory (abstract, line 23-26, line 86) but the main measurement used in figure 1 is the correlation of the velocity profile (which actually shows a strong increase in speed during learning). This is problematic for several reasons:– An inattentive reader may think that such that the y-axis label "reach correlation" in panels e, f, and g of figure 1 is referring to the correlations of the trajectories shown in c.

We thank the reviewers for this comment. – we have renamed reach correlation to “velocity profile correlation” to decrease this potential confusion. Please also see next point.

– This choice raises the question of why the authors quantified behavior by looking at the consistency of the speed profile rather than directly through trajectories correlation. Looking at the traces of the trajectories in Figure 1C, the improvement in trajectory consistency is not clear (day 2 seems to be more consistent than day 8). If the improvement in trajectory consistency is less pronounced than the increase in fast speed consistency, the authors should make significant modifications in the way they present their behavioral data and acknowledge that their physiological changes could explain either trajectory consistency, increased (fast) speed consistency, increase speed or a mixture of the three.

We thank the reviewers for the opportunity to expand on our methods. We have revised our manuscript to include a more elaborate and explicit rational behind choosing our learning metric. We will summarize our reasoning for quantifying the consistency of the velocity profile of the reach, rather than the spatial trajectory, below:

First, the spatial trajectory is constrained by both the task (the rat must reach through a small window in their box to a pellet location held constant) and the fact that it does not inherently capture temporal components. Over the course of learning, we observed that the spatial trajectory does not capture the main changes in reach strategy, which tends to be the speed and consistency at which the reaching trajectory is traversed. As our goal was to capture learning- (and sleep-) related changes in behavior, we found measuring changes in the velocity profile to best capture learning.

Second, we noted that animals often vary in the initial location of their reaching action, thus making day-to-day differences in spatial trajectory correlation difficult to interpret. Using the velocity profile and time locking trials to the moment when the rats interact with the pellet allows us to capture the consistency of the reaching velocity while approaching the pellet, while largely ignoring differences in starting location.

Third, one of the main changes that we observe during the learning of our task is the “smooth binding” of sub-movements – i.e., reaching towards the pellet, grasping the pellet, and retracting the pellet to the mouth – that were initially distinct. While all these movements still occur and traverse roughly the same path as in early training, with practice these movements are combined into a single, smooth action. The velocity profile appears to best capture this phenomenon.

– Increase in speed consistency does not necessarily imply an increase in speed. The striking increase in reaching speed with learning (Figure 1D) is just shown for one animal. The authors should show it for their 6 animals. If the 6 animals show an increase in movement speed this is a point that should be discussed throughout the manuscript, especially in light of the many works linking dorsal striatum and movement speed.

We have included a supplementary figure (Figure 1 —figure supplement 3) showing mean single-trial peak velocity across learning for all animals.

– In figure 1e, the authors showed in grey the individual speed profile correlation of 6 rats and the mean+sem. Something is quite wrong with the mean trace. In the first days, the mean should be much higher, close to 0.8. Currently, its first value is between the 5th and 6ht values. Second, on a statistical point of view, using mean and SEM is meaningless when n=6. The median would make more sense and there is no need for error bars as the entire dataset is shown. Once the group representation is corrected, taking into account that the y-axis in e) is cut at 0.6, it will be clear that the increase in speed profile consistency is far from impressive.

We apologize for this error and sincerely thank the reviewers for identifying it – we significantly edited Figure 1 and included a supplementary figure (Figure 1 —figure supplement 2) that displays all individual animal traces.

– To demonstrate that the behavior is automatic, in the new pellet location task, the authors used trajectory correlation to quantify behavior, that is, a different metric than in figure 1. This inconsistency in behavioral metrics across two related figures (Figure 1 and Figure S1) raises the question of whether the lack of behavioral change shown in Figure S1 (which the authors use to claim automaticity) is robust when looking at speed (either absolute speed or consistency). Added to the fact that this experiment was only performed on two animals this part was really not convincing. Anyway, the authors should also examine whether movement speed is affected by the relocation of the pellet.

We appreciate the reviewer’s concern and perspective on our measure of automaticity. In an effort to keep the revisions of reasonable scope, we agree that a compelling demonstration of this definition of automaticity is unachievable as we did not carry out the test of automaticity in all of our learning cohort and would therefore need to perform a new set of recordings to add further animals. Instead, we have focused on the long-term emergence and stabilization of a skilled action and its link to offline corticostriatal plasticity.

2. Validate LFP recordingsTo study M1 -DLS functional coupling, the authors, in some of their analyses, used striatal LFP to compute M1 and DLS coherency. When introducing their result section, the authors stated that "within the corticostriatal network, theta coherence (4-8Hz) has been previously shown to reflect coordinated population spiking activity8,9,34 " (Line 128). The authors should mention recent works showing major volume conduction in the striatum in this frequency range (Lalla et al., 2017) and in the γ range (Carmichael et al., 2017), as the authors seem to be aware of this potential confound (in Lemke et al. NN, 2019, relevant works are cited). Sleep rhythms are well known to be controlled by the thalamocortical systems and the LFP' sources to be in the cortex (Kandel and Buzsaki 1997). The lack of organization of the input on striatal neurons along with their radial somatodendritic shape makes the striatum a poor candidate to generate fields that can summate and be recorded extracellularly. Because the striatal recording sites in the present study are located just below the cortex it is very likely that most, if not all the striatal LFPs is volume-conducted from neighboring cortical sources. The authors said they used common-average referencing to limit volume conduction but this will not fix the problem. Indeed, subtracting two oscillatory signals with the same phase and frequency, but different amplitude (as it can happen in the striatum due to the passive attenuating effects of the brain tissue on LFPs) will result in an oscillatory signal with a preserved rhythmicity. The authors should look carefully at the work of Carmichael and collaborator (2017). In this study, the authors showed that striatal LFPs amplitude were slowly decreasing as the striatal electrodes were further away from the cortex. Carmichael also showed that striatal spiking modulation by striatal LFPs is not a criterion for local generation of the field (all it shows is that striatal units are modulated by cortical rhythms which is expected as the cortex provides strong excitation to the striatal neurons). If the authors want to make a claim about striatal LFPs being local then they need to show that their striatal LFPs (referenced against their cerebral screw) are not progressively decreasing away from the cortex. It is an important issue that cannot be ignored by the authors.

We appreciate the reviewer’s concern regarding volume-conducted LFP signals and have carried out the following revisions to address this important point.

First, we have revised the text and included citations to outline the issue of volume conduction in the striatum. An important point about our current method is that the common average referencing is performed in DLS and M1 *separately*, not across M1 and DLS signals together, thus pre-empting the issue cited of subtracting different magnitude oscillations. Moreover, in this work, we use increases in M1-DLS LFP coherence to argue that M1 and DLS become more functionally coupled during offline, rather than online, periods. Our revisions therefore seek to provide compelling evidence that M1-DLS LFP coherence reflects functional connectivity, rather than making a direct claim about the local and non-local components of DLS LFP.

Second, we have added a new supplemental figure (Figure 2 —figure supplement 1) showing that volume-conducted LFP signals are not a major contributor to 4-8Hz LFP coherence. This figure displays the non-zero phase difference between 4-8Hz LFP signals in M1 and DLS during NREM sleep. If 4-8Hz LFP coherence is simply reflecting volume conducted signals, then we would have observed 4-8Hz LFP signals in M1 and DLS that have zero-phase lag, consistent with volume conduction. However, a non-zero phase lag in M1 and DLS LFP signals between 4-8Hz during NREM is incompatible with volume conduction. We have previously used this method to show that reach related LFP signals across M1 and DLS have a phase lag consistent with the conduction and synaptic delay between M1 and DLS, and inconsistent with volume conduction (Lemke, et al., 2019).

Third, we added a new supplemental figure that links changes in M1-DLS LFP coherence to a separate measure of M1-DLS functional connectivity that is independent of DLS LFP: the phase-locking of DLS units to M1 4-8Hz LFP signals. Importantly, this separate metric does not require a local generator of striatal LFP and is therefore not susceptible to volume conduction. We show that M1 electrodes with high LFP coherence with DLS electrodes also entrain DLS units to a greater degree than M1 electrodes with low coherence, providing evidence for M1-DLS LFP coherence as a measure of functional connectivity.

Lastly, while we do agree that the reviewer-proposed method would provide compelling evidence if we observe no change in LFP coherence with increasing distance from cortex, we believe such decreases may also occur due to changes in corticostriatal projections patterns across the dorsal-ventral axis of the striatum. Moreover, as outlined above, our goal is not to argue that DLS has a local field in principle, but that cortical inputs to striatum change with learning/sleep. We believe that examining both phase differences and a spike based complementary approach provide support for this.

– Monosynaptically connected pairs and cross-correlograms (CCGs). The authors claim to identify monosynaptically connected neuron pairs from CCGs. Previous works (see for instance Bartho et al. 2004) have shown extremely sharp peaks in CCG of putatively connected neurons in the cortex and hippocampus but something is clearly different in the CCGs shown here. Indeed, it is striking that the CCG peaks shown are very smoothed (it is more a wave than a peak). In fact, this wave crosses the center of CCG and seems significant in the positive time bins, suggesting that striatal neurons fire before cortical neurons, which makes little sense. It is surprising that the authors did not mention in their result section the potential alternative mechanisms explaining such "peaks" that spread until positive CCG values. Indeed, it is well known that there are alternative explanations for these observations, such as indirect polysynaptic partners as well as common ("third party") input or slower co-modulation (see Brody CD (1999) Correlations without synchrony, and several papers by Asohan Amarasingham and colleagues). Amarasingham et al. (2012), J. Neurophysiology, talks about interval jitter, why it's relevant to separating fast from slow comodulation and explains the history of these problems. In this regard, the shuffling method briefly mentioned by the authors in the method section is unclear and does not seem to address properly the issue of separating fast and slow comodulation (see Amarasingham 2012).

We thank the reviewers for their critical points regarding the cross-correlation analyses. We have made significant revision to our manuscript on these analyses, utilizing a more conservative “basic jitter” method for detecting significant short-latency spike timing relationships. We have made major revisions to the main and supplemental figures reflecting this change. Additionally, we have included a more explicit discussion about the shape of our CCGs in contrast to what has been reported in cortex.

– While M1 RM significance passed the arbitrary α of 0.05, it is a dramatically weaker effect than that seen in DLS. I would ask the authors please comment on this difference. Additionally, both Δ and SO strongly modulate M1, and **to a greater extent than any of the other effects mentioned in the text!** Please point this out to your readers and offer an interpretation. Do these results argue that in M1 – unlike DLS- the spindles not the main contributor? Or if nothing else, that the modulation is non-specific? (I think the former. this may actually help the authors in their attempts to dissociate m1 from dls sleep processes). The spindle story is great, but these results change the interpretation should not be buried.

We appreciate the reviewers raising this point. As we have significantly changed our method for detecting connected pairs of M1 and DLS units (see above point), we did not end up with enough coupled pairs to make robust claims about differences in NREM rhythm modulation between coupled pairs and non-coupled pairs. Therefore, in an effort to focus on relevant analyses for the story, we have removed these analyses. In addition, we agree about the importance of δ waves and slow oscillations in cortex and, although we cannot make strong claims about corticostriatal processing (besides that DLS units are modulated to these rhythms as shown in Figure 5), we have included further discussion of these rhythm in the main text and discussion.

3. Neural TrajectoriesThere were 3 major concerns raised (1)how the trajectories are composed, (2) the validity of the predicted trajectory and (3) the comparison of the trajectories:1) The pc plots represent trial-averaged activity. Although no reach dynamics are given in this paper, a very similar study by the same authors shows that early on, the variability in duration of the outward, and outward+grasping, are very high compared to late in learning. Whatever 'reach signal' is present, this variability will cause the trial-averaged values to be greatly reduced. It is recommended that the authors address this by: (a) time-warping individual trials to overcome this and (b) doing this for only the outward trajectory, as their previous paper shows that so much of the change in duration due to learning is the result of improved grasping ( e.g the difference between PT and RO in Lemke 2019).

We thank the reviewers for the opportunity to clarify this analysis. While we present trial-averaged neural trajectories in Figure 3, the PCA is performed on concatenated single trial spiking data (not trial averages), and the predictions of DLS neural trajectories are also performed on a single-trial basis. We have clarified this in the text. In addition, we have included a comparison of DLS spiking modulation from early to late training days, as well as a comparison of variance explained by top PCs from early to late training days, to argue that changes in the ability to predict DLS activity from M1 activity is not solely attributable to local learning-related changes in DLS. Furthermore, we have also confirmed that we see similar results when restricting our results to the outward reaching action, although with increased variability as we are predicting ½ of the data in the original analysis:

2) It is also unclear if the authors are trying to say in Figure 2 "Offline increases in functional connectivity predict the emergence of low-dimensional cross area neural dynamics during behavior" The authors should disambiguate cross area dynamics from within area dynamics. Previous work has shown that there is little to no goal-related movement activity early on in training. More importantly, the authors themselves show this in their 2019 paper that although there is some Day 1 m1 spiking modulation related to the reach, the psth for striatum is flat psth (previous comments on variability of timing still apply). They also show some reach-aligned power increase in 4-8 Hz in M1, but nothing in striatum. Given the small dynamics(?) that might be in the striatum, it is actually impressive that there is as much similarity between DLS and predict as there is. Can the authors show that this is a lack of a cross-area relationship, not just a lack of signal – which would preclude even the possibility of that relationship. Potentially they could scale with the power of the signal – amplitude or % explained variability of the first PC's, or just plot avg dms modulation in b. Otherwise, the cross area dynamics argument is not compelling. This same problematic facet comes into play later when determining reach modulation (which was done across all days).

We agree with the reviewers that this is an important point. Please see above reply regarding the two new analyses we have included to address the potential confound of a ‘reach signal’ lacking from DLS on early training days. This is consistent with our previously reported results (Lemke et al., 2019, Supplemental figure 7), where we see no large change in percentage of task-related units in M1 or DLS with learning

3) Finally, how do you quantify quality of prediction? "The predictive ability of these models was assessed by calculating the correlation between the actual neural trajectories and the predicted trajectories." This explanation lacks vital details about the comparison and the model itself. Moreover, these are time-varying 2-D vectors, where one offset or delay early on can propagate throughout the trajectory, even though the trajectories would otherwise be identical. Much work has been done on this in recent years, as this is a difficult question to tackle. Euclidean distances of a time-warped distribution or point distribution models might help. Regardless, more detail is needed.

We have expanded our explanation of these analyses in the methods to address these points.

4. Clarify data presentation:– Figure 2c – The authors show an interesting temporal dynamic here, and there are hints that online DEcorrelation may contribute to learning (for which there is some evidence) as well as offline increases in coherence. This may be just a happenstance of this specific example, but 2c makes it clear that showing the trends across time, rather than in sum as in Figure 2d, would be helpful. Could the authors please add a figure like 2c, but across all 6 animals? It would be nice to see this possibility of online changes at least briefly discussed as well.

We thank the reviewers for bringing up this interesting point. We have included a supplementary figure that shows the mean change (across electrodes) for individual animals.

– Figure 4 sup2, etc – The cited conduction delay between m1 and DS is very specific: 5-7ms (and with a decrease beyond baseline in the 1-3ms range). The authors should redo their analysis using 1-10ms to 5-10ms, or provide a clearly compelling rationale for including 1ms latencies.Moreover, it is unclear why the x scale is +/- 75ms? this seems quite large and belies the bigger question of why the distributions are so large, especially for data that should be normalized (subtracting out baseline correlations, and thus should be sharper). The largely symmetric and broad increase, extending before 0 lag, is worrisome. If it is a function sleep vs wake, can connectivity be determined only during wake?

To clarify our method, although we cite a specific delay, we use a wider window as suggested for analysis (1-15ms), we have clarified this in the text. Regarding the broad width of the cross correlations, please see above point regarding CCGs.

– AP5 effects:The presentation and description of the results are confusing and incomplete. The experimental design could not be interpreted by reading the result section or looking at the figure 1 (panel f and g). In the method section, the authors wrote that " In the first five days of training, we infused three rats with AP5 and three rats with saline, for the second five days, we switched the infusion, i.e., animals that received AP5 in the first five days, received saline for the second five days, and vice-versa". First, this sentence should be included in the main text. Second, why did the authors show only one animal in figure 1f ? they should show the six rats.

We thank the reviewers for pointing out this confusion. We have included a supplementary figure (Figure 1 —figure supplement 4) with all individual animal curves.

– I could not make sense of the analytical logic in panel g. The authors correlated the trial by trial speed profiles during training with the average speed profile at the end of training, saline, or AP5 sessions. During "control" training (saline and learning cohorts), as the behavior changes, there should be lower correlation at the beginning of training and higher correlation at the end. In addition when the speed profile does not change (early AP5 or late learning cohort) there could be higher correlations. It is not clear to me how this measurement can be useful as it may vary a lot during learning and could also be different depending on the type of behavioral plateau.

We have revised Figure 1 panel g to show across animal differences in total correlation change.

– Also it is unclear whether the effect of AP5 on day-to-day correlation should be the same when injected since day1 or after 5 days of practice under saline condition. The data shown for one animal seems to show a clear decrease in speed profile correlation during AP5. What do the authors make of such effect ? It seems to indicate that AP5 not only block automatization but also impair performance. Is a similar effect observed when AP5 given after 5 days of saline ? The effect of AP5 on movement speed should be shown.

We have included a supplementary figure with all individual animal curves.

– Histology is absent. At a minimum, ascertaining how well layer 5 was targeted across animals should be included.

In this study we recorded physiology signals using large, combined arrays that targeted M1 and DLS, resulting in difficulty removing the arrays without damage to M1. This impeded our attempt to precisely localize the depth of electrode tips with histology. Our method to standardize depth of electrode insertion is to use precise stereotactic insertion of electrodes to 1.5mm from the brain surface, targeting layer 5 of motor cortex. In lieu of precise localization of electrode tips, we have performed gross histology and have confirmed that we are targeting M1 and DLS, we have included this in a supplemental figure.

– Reviewers were confused by sleep nomenclature:Figure 2a – it was not obvious that sleep 3 = next day's sleep 1. can you include something to the effect of "sleep3 = day n+1's sleep 1" just to help make it perfectly clear to the readers?Another commented that "pre-sleep" and "post-sleep" nomenclature very confusing. These aren't used to define time periods before and after a sleeping episode, but instead are periods of "sleep" (although there can be wake states mixed in) before and after training. "Pre-training sleep" and "post-training sleep" would be much more clear.

We appreciate the reviewer’s identification of this confusion and have edited this figure to clarify.

5. Reach modulation across days– Reach Modulation appears to be determined across all days? This assumes that each unit is maintained across 8 days- evidence supporting this needs to be provided. More importantly though, the potential for this to change the DLS results is remarkable. As previously stated, the authors showed very little reach related activity on day 1. Why not determine these values for day 1-2 vs day 7-8? or better yet, as somewhat suggested by the authors, can they predict what neurons will become RM over time?

We appreciate the reviewer’s identification of this confusion and have edited this figure to clarify.

– It seems that the day-to-day correlation measurement allows the authors to have more data points per rat. However, these measurements for a given animal are not independent. Unfortunately, pseudoreplication is a major issue in neuroscience and behavioral studies more broadly (S.E Lazic's "The problem of pseudoreplication in neuroscientific studies: is it affecting your analysis?" BMC Neuroscience, 2010, highlights some things to worry about in this regard). The authors should avoid this in this figure and other neurophysiological analyses.

We thank the reviewer for raising this point. We have edited this panel to compare total change in correlation value across days per animal (one value per animal).

6. NREM sleep– It would be interesting to know how many times the animals entered NREM sleep between the post-training sleep day N and pre-sleep of subsequent day N+1. Are these between day changes depended upon the total amount of sleep that the animals got?

We agree wholeheartedly with the reviewers that this is important and interesting information (and we hope to collect such information in the future!), but unfortunately, due to the challenge of such continuous recordings, we do not currently monitor animals both during post-training and overnight.

– pg6 line 192. In order to ascertain that NREM > others, it would be useful to see this relationship in unconnected pairs. e.g. control for the possibility that NREM is always highest.

We thank the reviewers for this point – we have completely redone our analysis of spike timing relationships between M1 and DLS neurons using a more conservative approach – this resulted in ~2.6% of pairs with a significant short-latency relationship, therefore it may be difficult to interpret this relationship in “non-connected” pairs. However, we have also included discussion in the text regarding the fact that there are clearly relationships in activity between M1 and DLS at longer time scales, however it is unclear how to interpret these timescales since they are likely resulting from slower common fluctuations.

– Since all of the data is from a specific subsection of these offline states (i.e. – NREM sleep) it would be good to discuss their findings within the context of what is known/unknown about plasticity processes during offline wake states or other sleep periods.

We share the reviewer’s interest in other offline states and agree that an interplay among activity patterns across different behavioral states is likely critical. We now include this point in the discussion.

7. Topics to clarify/discuss– Page 7 Line 234 – "In contrast, DLS units did not increase in modulation during either δ waves (Supplemental Figure 5) or slow oscillations (Supplemental Figure 6) after training." This statement is only partially correct. I applaud the authors giving, and showing, the same treatment for all 3 types of data. Many would gloss over this. DLS nonRM weak input demonstrated a stronger effect than M1 RM, and SO demonstrated a trend. Some will find this juxtaposition – task related-spindle, unrelated-other to be informative, particularly as information itself can be more strongly represented through increasing the signal or increasing the discernability from not-signal. There is considerable work, including from the authors, showing the importance of other elements of sleep. Throughout the paper, SO and δ aren't negligible contributors. Rather than glaze over this for the purpose of a throughline, please expand upon this for a more complete view.

We appreciate the reviewer’s note and have expanded the text on the potential roles or interactions between sleep spindles, slow oscillations, and δ waves in the discussion.

– The discussion is somewhat lacking, being just a summary of the findings. In particular, I would like to see the authors put these results into context with the broader models of Tononi (SHY) and their own work, e.g. Kim 2019 wherein they focus on SO and δ in a very similar task – but which are largely thrown under the bus here.

We appreciate the reviewer’s note and have expanded the text on the potential roles or interactions between sleep spindles, slow oscillations, and δ waves in the discussion.

– It is unclear why "automaticity" is a beneficial outcome here for this task. Does it result in greater pellet retrieval success (no data presented on this point) or does it free up or allow reorienting of cognitive resources (this is indirectly alluded to in the Discussion)?

In line with our above response, it is difficult to interpret automaticity in this task and have therefore focused on the emergence of a stable skilled action. It is certainly possible that automaticity is beneficial in this task because it reduces cognitive load and therefore makes the action more “cognitively efficient”. However, we have not designed the experiment to capture this “cognitive” aspect of automaticity.

– The argument structure in the Intro has a few holes in it.a) Specifically the paragraph covering lines 42-53. "Currently, our understanding of how sleep impacts distributed brain networks is largely derived from the systems consolidation theory, where it has been shown that coordinated activity patterns across hippocampus and cortex lead to the formation of stable long-term memories in cortex that do not require the hippocampus 23-25. Notably, whether sleep impacts the connectivity across hippocampus and cortex has not been established. Therefore, one possibility is that, in the network, we similarly observe coordinated cross-area activity patterns during sleep but do not find evidence for the modification of corticostriatal connectivity during offline periods. Alternatively, it is possible that we find evidence that cross-area activity patterns during sleep modify the connectivity between cortex and striatum and impact network activity during subsequent behavior."b) The first two sentences describe the role of sleep in systems consolidation theory and "coordinated activity patterns" and "connectivity" across hippocampus and cortex. The third and fourth sentences then jump to two alternative speculations about "the network" (which is undefined), and the role of sleep in modifying corticostriatal connectivity patterns. It is not clear how these two things are linked. This link should be explicitly stated.

We thank the reviewer for pointing out our omission of the word “corticostriatal” before network and have revised the introduction to address these concerns.

– Offline is used in two different ways here. Offline behavioral states include any sleeping or waking state outside of training. Then there are online (between pre- and post-sleep on the same day) and offline (between post-sleep of day N and pre-sleep of subsequent day N+1) changes in neural activity between these offline behavioral states.

We note this potential area of confusion and have clarified our use of offline terminology.

– The authors need to better define what they mean by slow oscillations (Figure 7, Figure 4e). Cortical slow oscillations are classically described as regular up and down states transition, with sleep spindles occurring during the up states (Buzsaki, rhythms of the brain p 197). When the authors look at the spiking modulation by slow oscillations and spindles (Figure 4e) or the temporal proximity between slow oscillations and spindles (Figure 7), I doubt the authors are referring to this classical up/down slow oscillations. Are the slow oscillations the authors refer to equivalent to K complexes? If yes, this could be mentioned.

We agree that the use of slow oscillations can differ across the sleep literature and have included a supplemental figure that outlines our specific method to detect slow oscillations vs. δ waves vs. sleep spindles.

[Editors' note: further revisions were suggested prior to acceptance, as described below.]

The manuscript has been improved but there are some remaining issues that need to be addressed, as outlined below:1. A graph showing the effects of AP5 on speed, for all 6 mice should be shown in the main figure in a single plot with a fixed scale. All of Reviewer #3's requests for this figure should be included, including statistics.

This is now included as Figure 1e. Please see discussion below regarding additional aspects of Reviewer #3’s comments.

2. This figure is requested to address potential confounds of effects on vigor that are currently interpreted as effects on motor learning. Depending on how conclusive the effects of AP5 on speed are, the authors might need to revise their language throughout the paper, if an effect of movement vigor cannot be rules out.

We have included this as Figure 1 – Supplement 5. We did not find a significant effect on single trial peak reaching velocity. This suggests that these two aspects of behavior may be distinctly regulated. Please also see discussion below.

3. A paragraph should be added to the Discussion explaining the potential confounds of effect on movement vigor on the study's results.

Please see the paragraph starting on line 375.

Although these points came from Reviewer 3, in consultation the other Reviewers agreed these points above are important to address.Reviewer #3:The authors have made a significant effort to reply to most of my comments and thoses of the other reviewers. However one of the major points that I had asked has not been addressed and another related one is not addressed properly. Those points were critical for the authors claim that "our results provide evidence that sleep shapes cross-area coupling required for skill learning" (last sentence abstract, maybe a "is" is missing?).Specifically I asked the authors to report the effect of AP5 injection on movement speed. They responded that they provide the effect of AP5 on the speed profile correlation but clearly this is not what I asked. I don't think the authors misunderstood me because in reply to another comment about movement speed (my original comment :"-Increase in speed consistency does not necessarily imply an increase in speed. The striking increase in reaching speed with learning (Figure 1D) is just shown for one animal. The authors should show it for their 6 animals. If the 6 animals show an increase in movement speed this is a point that should be discussed throughout the manuscript, especially in light of the many works linking dorsal striatum and movement speed"), the authors did provide in their supplementary figure 3, plots showing that movement speed increases in the 6 animals during training. Thus it is unclear why the authors did not show the effect of AP5 on movement speed.My request is really fundamental in regard of the main claim of the author (requirement for skill learning) because it is possible that AP5 decreases not just the trial-by-trial speed profile correlation (what the authors use as a definition for skill) but also the general speed of movements which is not necessarily related to skill learning but could have a motivational origin (see work of the Galea lab). If this was the case, the authors would need to seriously consider that the changes in corticostriatal connectivity could primarily reflect altered vigor or motor motivation which would be in agreement with several works in the field (Rob Turner lab, Dudman lab, Robbe lab, or even ideas on motor motivation by Josh Berke, one of the authors of this study).This point is even more important as the authors, in the introduction or discussion, tend to write definitive sentences on a well-demonstrated role of cortico-striatal connection in motor skills while most of the references cited do not disambiguate the vigor or motivation confound (e.g., Kupferschmidt et al., Dang et al., Costa et al., Yin et al. …). This is quite misleading.Moreover, when looking at the result of the AP5 experiment on the speed profile correlation across the 6 animals (supp Figure 4), the results are far from convincing. There are only 3 animals in each condition (3 in AP5-Saline and 3 in Saline AP5) and in each of the conditions, the result is not clear in at least one animal. There are also weird day by day drastic changes that make these results not so reliable. Moreover, the authors keep changing the y axis scale in a way that makes small improvements look big. Ethically speaking this is not appropriate, especially in such a journal as eLife. Neither is appropriate to put in the main figures 1d and 1e the best example out of 6 animals. The authors should integrate into these panels the results of the other animals. In addition to being fairer to the data, this would remove two supplementary figures.Thus, the main figure 1 should show 1) the effect of AP5 on speed profile correlation on the 6 animals in the same graph with the same scale (Figure 1e); 2) the effect of AP5 on max speed on the 6 animals (new small panel) and the fig1d for all the animals. In addition, the statistics to examine the effect of AP5 should be done using paired comparison of the correlation and speed values not on their change. The usage of the changes in panel 1f is unfair because it is not clear what should be the effect of AP5 after saline (decrease or plateau?) I am afraid that the authors designed their statistical test in a way that favored their hypothesis.In conclusion, while I do think this paper should be published in eLife, I am convinced that a fairer presentation and analyses of key experiments in figure 1 are required. Clearly, the effect of AP5 on skill learning is not as clear-cut as stated by the authors (and choosing to only show in the main figure the single best animal is not appropriate). In addition, an effect on movement speed is still highly likely (based on one animal) and should be reported for the 6 animals. Thus the main behavioral function of the interesting neurophysiological changes could be related to vigor/motor motivation, not skill learning. All this critical information should not be hidden but rather openly disclosed and discussed such as the reader can make up its mind.

Challenges in measuring movement vigor in the reach-to-grasp task:

We wish to emphasize the challenges in determining movement vigor in the current work, specifically with the reach-to-grasp (R2G) task. As we are sure the reviewers are aware, published work on movement vigor often utilizes highly constrained tasks in which mice are head-fixed, which reduces variability in body position, and are gripping a joystick that only can move in a single dimension (these joysticks also typically have a “resetting” force so the only movement the mouse needs to make is outward). Such a preparation is ideal for carefully measuring specifically the vigor of a movement which is constrained to be very similar (in terms of muscle activation patterns) across different animals.

Our task is quite the opposite! As made clear by the seminal work of Ian Wishaw and colleagues, the R2G is a sequence learning task; the animals appear to learn to correctly order and time “sub-movements” (e.g., paw plant, reach, grasp, retract) in order to solve the task. For rodents, this movement is typically so variable in early stages that they rarely successfully grasp the pellet in the early stages. Thus, it is not obvious that a simple gain change is sufficient to drive learning.

Consistent with this, there are numerous reports of cortical plasticity associated with R2G learning (e.g. Kleim et al, J Neurophys 1998; Xu et al., Nature 2009). By design, this task is unconstrained, so as to allow the animals to explore the many degrees of freedom associated with skill learning and each animal is free to develop a unique strategy. This is the reason why we use a higher-dimensional readout of behavior, the correlation between velocity reaching profiles, rather than reduced representations such as maximum velocity. In fact, the nature of our task is not well-equipped to measure and make strong claims about movement vigor because, while we do report maximum velocity of the outward reaching movement, this value not only depends on the “vigor” of movement, but also the starting body position of the animal which is not constrained (for example an animal reaching straight-on toward the pellet vs. an animal reaching from an angle of 15 or 30 degrees would require a different set of muscle activations), as well as the distance of the reach.

Moreover, while maximum velocity is one aspect of the reaching action, it is likely one of many features that change with learning. For example, perhaps the most important aspect of “movement vigor” is the grasp of the pellet – a force we do not measure in the current work. All that to say is that we do completely agree that a change in movement vigor may play an important part in learning. However, as noted above, a simple “gain” change in movement vigor would not likely explain the changes we see in the consistency of the reaching velocity profile.

We do, of course, seek to give a complete and fair assessment of changes in movement velocity and how they might relate to skill learning: during the first revision we included maximum velocity for all animals on all days of learning, we have now also added the correlation between maximum velocity and consistency of reaching velocity profile in the learning cohort (Figure 1 – Figure 1 Supplement 3) as well as a figure displaying the changes in maximum velocity for AP5/saline animals. From the correlation between maximum velocity and consistency, we see that the consistency and maximum velocity of reaching movements are linked. As mentioned above, we don’t interpret this as evidence that maximum velocity fully explains the change in stability (R^2^ value = 0.13), but rather that it is one of the features that changes as a part of learning. Furthermore, we do not find a significant change in maximum velocity comparing AP5 and saline (Figure 1 – Figure 1 Supplement 5), suggesting these two aspects of behavior may be distinctly regulated. As requested, we have also expanded on the control of movement vigor by the basal ganglia in the discussion.

Inter-animal variability in the reach-to-grasp task:

We seek to emphasize another critical aspect of our task in response to the reviewers concern about differences in the magnitude of reach-profile correlation values across animals. For relatively “constrained” motor tasks, e.g., joystick press, inter-animal variability is often much lower in compared to relatively “unconstrained” tasks, e.g., reachto-grasp task. Such inter-animal variability has been studied in the reach-to-grasp task; in the well-regarded work by Nitz & Kargo, J. Neuroscience, 2003 - they examined the different “strategies” that animals utilized to learn the reach-to-grasp task (from their paper: “Skill improvement was associated with both motor pattern selection and pattern tuning. One group of animals (3 of 11) appeared to switch between motor patterns underlying the reach portion of the task…. In contrast to the first group of animals, a second group (8 of 11) appeared mainly to tune a single starting motor pattern over time”). Consistent with their finding that different animals tend to undergo different “paths” to learning, we believe that an increase in correlation value from, for example, .4 to .6 vs. .8 to .9 may both reflect relevant aspects of motor learning, while the correlation magnitude value itself is not easily interpretable. However, as to not “hide” this variability, we have included all animal data in plots with fixed scales in Figure 1. We have no desire to hide data and appreciate this opportunity to be more transparent.

The reviewer’s also suggested that we change our comparison to do a “paired comparison of the correlation and speed values not on their change”. As discussed above however, we believe the change in correlation is much more interpretable, than the absolute value of the correlation values which vary between animals. Another issue with the proposed comparison is that, as the reviewers points out, in half of the animals AP5 infusions occur after five days of learning with saline infusions (and vice versa), thus the animals have already undergone learning and the correlation values are higher. When we planned our experimental method, we organized our experiment in this way to reduce the number of animals needed, with the plan to use a statistical approach that compared changes in correlation values rather than the raw values themselves. This was informed by the clear evidence of changes in correlation during our initial studies in normal learning. We wish to also note that our study design was done in a randomized blinded manner in order to add further rigor. The experiment is, of course, balanced such that the same number of animals receive saline infusions after AP5 infusions, as AP5 infusions after saline infusions. Thus if AP5 had no effect on learning, we would not expect to see a difference in correlation change, as we do.

[Editors' note: further revisions were suggested prior to acceptance, as described below.]

Reviewer #3:I asked the authors to show the data for the 6 animals in which they compared AP5 and Saline striatal injection on the speed correlation profile. The authors have managed to plot the data in a way in which it is impossible to know which rat is which (all the points are gray, no running lines between the points). They plotted the mean of these data which is meaningless from a statistical viewpoint (n=6).

– We replotted the data to connect the points of individual animals (Figure 1d and e).

– All individual animal plots (of behavioral and neural data) are also presented in the supplemental figures (Figure 1 Figure Supplement 2,3,4,5; Figure 3 Figure Supplement 1).

I also asked the authors to show on the same main figure the effect of AP5 on the max speed (to test the effect on vigor). The authors have put those results in a supplementary figure. The authors claim that AP5 has only an effect on the speed correlation profile not on max speed. But this is clearly due to their biased statistic in which they only look at the difference between the first and last sessions. Indeed in figure 1e AP5 only reduces the speed correlation on day 5 but there is no effect from day 1 to day4. Juxtaposing the effect of AP5/Saline on max speed (now Figure 1 S5) and speed correlation ( figure 1e bottom) clearly shows that the effects are strikingly similar. In fact, an unbiased treatment of the data (using the entire profile and permutation/bootstrapping) would probably show that the effect of AP5 on max speed is more pronounced than on speed correlation.

We expanded our treatment of the data to show this is not the case.

– We added a test of significance for these metrics using permutations/bootstrapping on day-to-day changes, in addition to the Whitney rank-sum test on total change in each animal (Figure 1; Figure 1 Figure Supplement 5).

– We expanded the data presentation to include plots of single trial peak velocity across days in individual animals and added histograms of day-to-day changes with AP5 vs. saline infusions for both single trial peak velocity and velocity profile correlation, which include mean day-to-day changes in each animal (Figure 1; Figure 1 Figure Supplement 4 and 5).

The authors seem to conclude that the AP5 effect is different on speed profile correlation and max speed because their statistical comparison with saline is significant in one case and insignificant in the other case.However, comparing p values is meaningless statistically. This is an important issue that has been subject to publication in highly visible journals (https://www.nature.com/articles/nn.2886).The authors should have compared the effect themselves which again should be done using permutation/bootstrapping on the entire profile (not last/first).

We do not seek to claim that AP5 is having an effect on a specific aspect of the reach-to grasp movement, but rather that post-training AP5 infusions impact skill learning, which we measure by looking at the emerging consistency of the velocity profile.

– We have revised the text to ensure we are not implying that AP5 is having a different or specific effect on our learning measure vs. single trial peak velocity. We certainly think reach velocity is a relevant feature of the learned action; however, it cannot capture all the changes involved in learning the reach-to-grasp skill (which involves an outward reach of the paw, a dexterous interaction with the pellet/grasping, and a retraction of the paw, all combined into a smooth and consistent skilled movement).

Thus I maintain that impartial analyses of the data cannot disentangle whether the DLS is primarily contributing to learning the accurate movement or contribute to the increased vigor which is driven by a motivational aspect. I am glad the authors acknowledge it in their discussion. However, most readers will go quickly through the title and abstract (and maybe introduction) and will probably cite this paper as additional evidence for a critical role of the striatum in motor skill learning which in my opinion is misleading.

We have added further evidence that physiological changes across the corticostriatal network covary with the emergence of a consistent velocity profile, beyond what is attributable to simply changes in peak velocity.

– We have included the computation of the partial correlation coefficient between mean corticostriatal LFP coherence during the pre-training period and the subsequent training period’s velocity profile correlation value, while controlling for the subsequent training period’s mean single-trial peak velocity (R value = 0.62, P value = 1*10-4; compared to R value = 0.73, P value = 5*10-10 when not controlling for single trial peak velocity). If, in contrast, we compute the relationship between corticostriatal functional connectivity and single-trial peak velocity, taking into account velocity profile correlation, we do not see a significant relationship (R value = 0.16, P value = 0.25).

– We also reemphasize that the goal of this work is not to disentangle the contributions of the striatum to movement vigor and movement consistency, rather to provide evidence that offline periods are relevant to skill learning because they help shape the corticostriatal network, which is known to be an important network related to learning. The reach-to grasp skill is a complex action made up of several movements that requires several effectors (to reach vs. grasp) that is not well-suited to isolating and studying the neural correlates of a specific movement feature such as reach vigor.